# Anharmonic quantum nuclear densities from full dimensional vibrational eigenfunctions with application to protonated glycine

Chiara Aieta [1], Marco Micciarelli[1], Gianluca Bertaina [1,2] & Michele Ceotto [1✉]

The interpretation of molecular vibrational spectroscopic signals in terms of atomic motion is essential to understand molecular mechanisms and for chemical characterization. The signals are usually assigned after harmonic normal mode analysis, even if molecular vibrations are known to be anharmonic. Here we obtain the quantum anharmonic vibrational eigenfunctions of the 11-atom protonated glycine molecule and we calculate the density distribution of its nuclei and its geometry parameters, for both the ground and the O-H stretch excited states, using our semiclassical method based on ab initio molecular dynamics trajectories. Our quantum mechanical results describe a molecule elongated and more flexible with respect to what previously thought. More importantly, our method is able to assign each spectral peak in vibrational spectroscopy by showing quantitatively how normal modes involving different functional groups cooperate to originate that spectroscopic signal. The method will possibly allow for a better rationalization of experimental spectroscopy.

[1] Dipartimento di Chimica, Università degli Studi di Milano, via C. Golgi 19, 20133 Milano, Italy. [2] Istituto Nazionale di Ricerca Metrologica, Strada delle Cacce 91, 10135 Torino, Italy. ✉email: michele.ceotto@unimi.it

Infrared and Raman vibrational spectroscopies are fundamental tools for chemists. In particular, the mid-infrared region is traditionally interpreted by assigning the characteristic vibration of functional groups to each band. These assignments have been inferred from experiments[1], and handbooks have been compiled with characteristic group frequencies[2,3]. This approach is indisputably useful for chemical analysis and very appealing to chemical intuition. However, spectral features correlates precisely with molecular structure only after appropriate quantum mechanical theoretical modeling. Despite the importance of vibrational spectroscopy, theoretical chemistry approaches may perform accurate simulations of quantum vibrational spectra only for few-atom systems[4–8].

The standard approach is normal-mode analysis. Normal modes are linearly independent atomic displacements that describe molecular vibrations in terms of harmonic forces. In other words, it is assumed that the interaction of the infrared (IR) radiation with a molecule results in a vibrational excitation confined to the small-amplitude regime. More specifically, the Hessian matrix, whose elements are the mass-weighted force constants, is diagonalized at the molecule equilibrium geometry. The link of spectral peaks to molecular motions is therefore provided by the correspondence between Hessian matrix eigenvalues, which are directly related to normal-mode frequencies, and the eigenvectors, which are sets of nuclear displacements. The normal-mode approximation justifies in part the traditional group frequency hypothesis. Indeed, it is true that for small and symmetric molecules the normal-mode analysis mostly results in motions of local groups. However, for larger non-symmetric molecular systems, the eigenvector displacements are necessarily spread all over the molecular structure, and the mode assignment in terms of functional group frequencies must be tentative[1,9,10]. Instead, chemical intuition is tempted to assert that a given vibrational normal mode is only weakly coupled to other modes in faraway parts of the molecule[11–13], so that distance-based truncation of couplings is a common view in chemistry. Consequently, many methods have been proposed to localize normal modes[12,14–18], to help with the interpretation of spectra in terms of chemically intuitive motions also for large systems.

A further problem is how to account for anharmonicity, which is particularly relevant when molecules undergo large-amplitude motions such as internal rotations. Classical molecular dynamics (MD) offers a way to go beyond the harmonic approximation. Specifically, trajectories that run on the actual nuclear potential energy surface (PES) at different energies may span regions far from the potential minimum surroundings. The Fourier transform of the velocity autocorrelation function from MD simulations yields at low temperature the same frequencies as the normal-mode analysis, but in addition, anharmonic frequency shifts can be estimated from high-temperature simulations[19,20]. Systematic methods have been reported to decompose the whole classical nuclear dynamics into an approximate sum of effective mode contributions, which correspond to the peaks in the spectrum[11,21–26].

Nevertheless, features such as zero-point energies (ZPEs), overtones, and tunneling splittings, require a quantum mechanical treatment of vibrational spectroscopy. In the quantum framework, spectral peaks correspond to transitions between vibrational states. Exact quantum methods for spectroscopy involve a solution of the nuclear Schrödinger equation to get the vibrational levels. However, peak assignment in quantum mechanics is more cumbersome than in classical mechanics. In fact, the dimensionality of internal coordinate eigenfunctions corresponding to eigenenergies for a (linear) molecule containing $N$ atoms is $N_v = 3N - 6(5)$. Therefore, differently from the classical picture which can be represented by classical trajectories, the normal-mode displacements are not directly interpreted as single point atom displacements in Cartesian

space. Moreover, as mentioned above[4–8], the exact solution of the quantum vibrational problem is in general limited to low-dimensional systems. Specific methods have been devised to get eigenenergies for higher molecular dimensions[27–36], but the eigenfunction estimates are usually accessible only for the ground state[31,37].

The multiple coherent time averaging semiclassical initial value representation (MC SCIVR)[38–41] can successfully simulate anharmonic vibrational spectra for large systems[42–46], even in extreme anharmonic cases[47–49]. MC SCIVR can be rigorously derived from the stationary phase approximation of the exact Feynman's path integral, and expresses the quantum time-evolution propagator in terms of a few classical trajectories, whose energy is close to the vibrational eigenvalues of the system (see Supplementary Methods). Moreover, MC SCIVR is also able to recover a reliable approximation of the vibrational ground and excited eigenstates by expanding the eigenfunction of each vibrational eigenvalue as a combination of harmonic eigenfunctions[50–52]. The combination coefficients are calculated by Fourier transforming the semiclassical approximate quantum mechanical time correlation function of each harmonic eigenfunction at the vibrational eigenvalue obtained from a preliminary MC SCIVR power spectrum calculation (see Supplementary Methods).

In this work, we introduce the calculation of one-nucleus marginal densities by Monte Carlo integration of the anharmonic eigenfunctions obtained with MC SCIVR simulations. Specifically, we employ the eigenfunction modulus square to weight a Monte Carlo Cartesian space sampling at each molecular conformation, and we obtain the final molecular nuclear density as a direct sum of each nucleus contribution[53], (see Supplementary Methods). This quantity is the nuclear analogue of electron density in Density Functional Theory for electronic structure calculations. The Cartesian coordinate space representation allows for the visualization of probability density isosurfaces in 3D, as for the nuclear ground state distributions obtained from Diffusion Monte Carlo calculations[31,37,54–58], and in a similar fashion as routinely done for electron density[59,60]. We are therefore able to represent with nuclear density differences the nuclear motion associated with each peak in vibrational spectra in three-dimensional (3D) space, and to visually spot couplings resulting in specific non-local nuclear density patterns in a quantum mechanical framework. In addition, we compute the anharmonic quantum density distributions of other observables, such as bond lengths, angles, and dihedrals using the same procedure.

## Results

**Protonated glycine vibrational eigenfunctions**. In this work, we calculate the 27-dimensional vibrational eigenfunctions and the one-nucleus marginal density for the protonated glycine (GlyH$^+$) molecule. Neutral glycine is the simplest amino-acid, and it has been extensively studied with both theoretical and experimental methods[43,61–63]. We choose to study its protonated form, because it is a typical product during infrared multiple photon dissociation (IRMPD) spectroscopy, one of the most effective experimental approaches for structural investigation of biomolecules[64,65]. Actually, in the gas phase, protonation is the dominant ionization pathway when analyzing peptides by mass spectrometry[64]. Among all possible vibrational eigenfunctions, we focus on the ground state and on the O–H stretch excited vibrational state with intake of one quantum of excitation. We focus on this excitation because the O–H stretching peak is usually very intense and it is employed as the reference to scale the calculated harmonic vibrational spectra to match the experimental ones[66].

Moreover, from the computational point of view, GlyH$^+$ is an ideal application to demonstrate the capabilities of MC SCIVR, since neither pre-computed PES nor exact quantum vibrational calculations have been reported with current state-of-the-art methods for this 11-atom molecule. We evolve the needed classical trajectories on-the-fly, and in this case we compute the electronic potential along the dynamics at the DFT-B3LYP/aug-cc-pVDZ level of theory (see the "Methods" section below).

In the harmonic case, the 27-dimensional vibrational Hamiltonian becomes separable in the normal-mode coordinates. Therefore, the harmonic wavefunctions are the direct product of 27 one-dimensional eigenfunctions, each one depending on a single normal-mode coordinate. The degree of excitation of each one-dimensional wavefunction defines the state. For instance, for the harmonic ground state, all the eigenfunctions in the product are the ground state solution of each separate one-dimensional Schrödinger equation with harmonic potential. Instead, the harmonic O–H stretch excited state is the same direct product as the ground state for the first 26 DOFs, but with the eigenfunction depending on the 27th normal-mode coordinate in the first excited state solution. For the anharmonic case, we represent the wavefunction with a combination of harmonic wavefunctions, as defined in Supplementary Eq. 14. The combination coefficients are obtained with the semiclassical procedure described in the Supplementary Methods[51]. The character of the anharmonic wavefunction is determined by considering the relative contribution of the different harmonic basis functions. In particular, the ground state wavefunction has the largest coefficient on the harmonic ground state (see Supplementary Table 1), while the O–H stretch excited state has the largest coefficient on the harmonic basis function with one quantum of excitation on the 27th component which depends on the 27th normal mode and whose displacement correspond to the stretching of the hydroxyl group (see Supplementary Table 2). However, in the latter case, we find other important contributions coming from other harmonic states that are mixing with the fundamental O–H stretch one. For example, mode 26 which is the asymmetric N–H stretching at fundamental frequency equal to 3504 cm$^{-1}$, mode 25, which is the symmetric N–H stretching at frequency 3445 cm$^{-1}$, and modes 23 and 22 which are the C–H symmetric and asymmetric stretching with harmonic frequencies equal to 3116 and 3105 cm$^{-1}$ respectively form combination states with low-frequency modes that have large coefficients in the anharmonic wavefunction expansion. Also, there is a contribution from the overtone of the O–H stretching with two quanta of excitation.

**Protonated glycine vibrational eigenvalues.** We start from the normal-mode analysis of GlyH$^+$ at the potential global minimum. Gas phase calculations have already been reported for this structure, in which the protonated amino group presumably forms an ionic intramolecular hydrogen bond with the carbonyl oxygen[67,68]. Our DFT optimized structure displays $C_s$ symmetry.

In a normal mode approach, we diagonalize the Hessian matrix at this geometry, and with the frequencies and harmonic zero point energy we build the stick power spectrum represented in green dashed lines in Fig. 1a. The O–H stretch is the highest frequency mode (3694 cm$^{-1}$) and it belongs to the $A'$ irreducible representation. More importantly for our discussion, the eigenvector displacements associated with this frequency result almost exclusively in the variation of the O–H distance, thus indicating a very localized O–H stretch motion, as pictorially highlighted in the ball-and-stick molecule reported on the inset in Fig. 1a. We detail the calculated Cartesian displacements along this normal mode coordinate in Supplementary Table 3 and Supplementary Fig. 7. To go beyond the harmonic approximation, we calculate the

semiclassical power spectrum with the red continuous line in the same panel, where we have singled out the ZPE and the fundamental O–H stretch peaks. The value of the ZPE energy from the global molecular minimum is well above 20,000 wavenumbers, which is a huge amount of energy in comparison with the harmonic vibrational level spacing. This quantum quantity cannot be grasped by any classical simulation and determines the physical behavior of the system as we will show in the next section. Here, we look at the influence of the harmonic approximation on the estimate of this key quantity. Figure 1b pictorially represents the semiclassical and harmonic vibrational energy level estimates. The red and the green arrows correspond respectively to the semiclassical and harmonic O–H stretch transition frequency. These frequencies can be directly compared with the peak positions of the experimental IR spectrum (black continuous line in Fig. 1a)[69]. The accuracy of our on-the-fly semiclassical approach outperforms the normal mode approach. The harmonic estimates of the ZPE and the O–H stretch do not provide an accurate value for the IR fundamental transition associated to the O–H stretch excitation. Actually, the harmonic transition frequency is significantly worse than the semiclassical prediction when compared to the experimental value, as shown in the inset Table of Fig. 1b. This confirms that the semiclassical treatment is remarkably able to describe precisely the actual spectral decomposition of the nuclear vibrational Hamiltonian, beyond the harmonic approximation.

**Protonated glycine nuclear densities.** In the previous paragraph we showed the GlyH$^+$ molecular system has a significant amount of ZPE, which is overlooked by classical approaches. Therefore, one may wonder to what extent the molecular average geometry at the ZPE level differs from the classical geometry at the bottom of the well (i.e., the position expectation value in the harmonic approximation). To answer this question, we calculate the full quantum distribution of the geometry parameters, using the nuclear densities both in the harmonic and anharmonic case, as described in details in the Supplementary Methods. Even if the ground state vibrational eigenfunction is mainly peaked around the classical minimum, we observe a slight elongation of almost all the bond distances and angle distortions in the anharmonic case with respect to the harmonic ones. Moreover, if we compare the anharmonic quantum nuclear densities with the corresponding harmonic ones, we can appreciate how the synergy of these small effects results in specific deviations of the global vibrational behavior. The actual nuclear harmonic ground state density is displayed in Supplementary Figs. 6a1–4, while Supplementary Figs. 6c1–4 shows the actual anharmonic ground state density. To clearly visualize the differences between these two nuclear densities, we report in Fig. 2b the difference between the anharmonic and the harmonic nuclear densities for the ground state, along with the atomic labeling to guide the discussion of the results in Fig. 2a.

The most relevant differences are found in the densities of light hydrogen atoms. Specifically, the three red lobes on the H1, H2, and H3 nuclei indicate an enrichment of anharmonic density in the inner part of the protonated amino group umbrella, which offsets the density depletion represented by the blue lobes on the outer part. This picture reveals that the effect of anharmonicity is to drive the ammonium protons closer together with respect to the harmonic picture (as highlighted in magenta in Fig. 2n, and as also confirmed by the angle distributions in Fig. 2h). The nuclear density shift on Hydrogen nuclei H4 and H5 determines an elongation of the C2-H4/5 bonds, as confirmed by the bond length distribution in Fig. 2f). Also, anharmonic effects lead to a

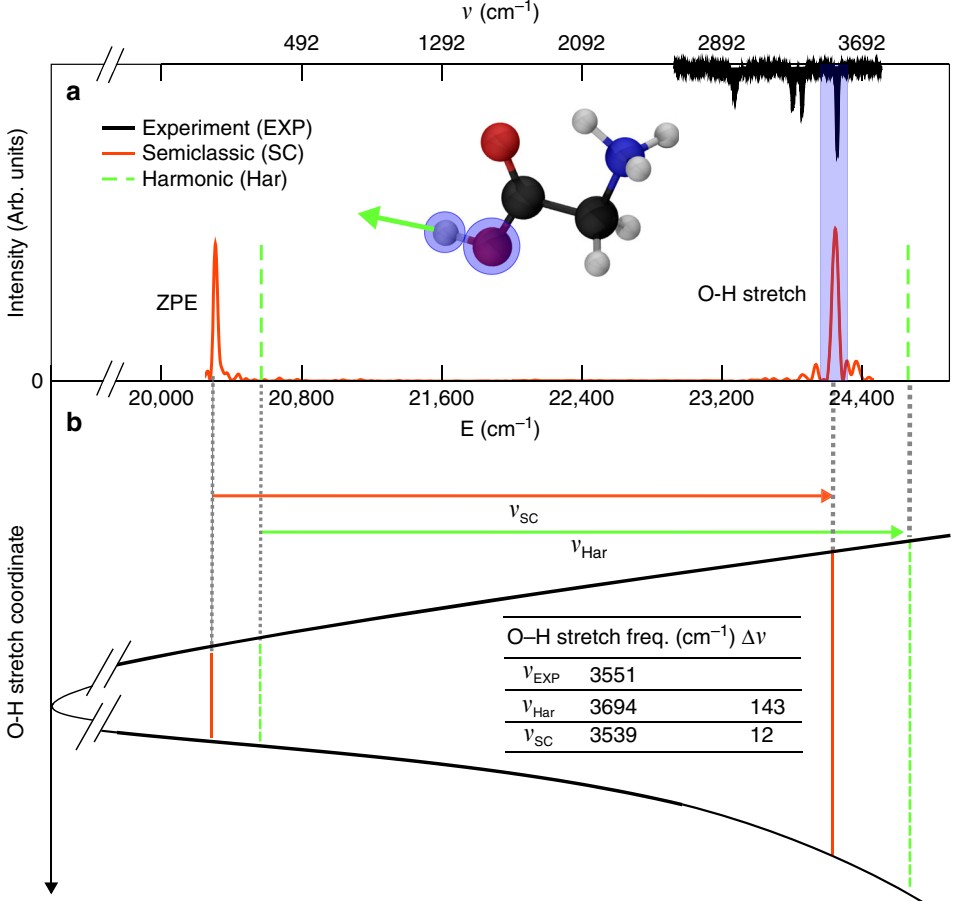

**Fig. 1 Vibrational power spectrum of the protonated glycine molecule. a** GlyH$^+$ vibrational spectrum. Red continuous line is the semiclassical power spectrum, and green dashed lines correspond to the stick spectrum in the harmonic normal mode approximation. The peak intensity of the power spectra has been scaled to match the semiclassical ZPE peak one. Black continuous line is the experimental IR absorption spectrum[69]. In the same panel, ball-and-stick molecular representation of GlyH$^+$ at the equilibrium $C_s$ geometry where the atoms involved in the O–H stretch normal mode displacements are highlighted. **b** Pictorial representation of the potential well along the O–H stretch elongation. Red and green lines are respectively the semiclassical and the harmonic ZPE and O–H stretch excitation vibrational levels. Red and green arrows are respectively the semiclassical and the harmonic normal mode O–H stretch fundamental excitation frequencies. These values are reported in the inset table together with their deviations from the experimental value ($\Delta\nu$).

displacement of the average position of the heavier O1, C1, C2, and N nuclei, thus rendering the backbone of the molecule slightly longer (as highlighted in green in Fig. 2n). This is confirmed by the bond-length distribution calculation, where the equilibrium distances C1–O1, C1–C2, and C2–N are longer for the anharmonic ground state wavefunction (see Fig. 2c, d, g). Also, the N–C2–C1–O1 anharmonic dihedral angle distribution of Fig. 2e is broader than the harmonic one. From bond-length distributions we also observe that both the N–H3 and the O2–H3 average distances are longer with anharmonicity inclusion (see Fig. 2i, l), hinting at a weaker O2–H3 hydrogen bond. However, the whole nuclear density picture of Fig. 2b shows that multiple local density deformations cooperate to the change of the H3 atom average position. Anharmonicity closes the protonated amine umbrella while opening the O2–C1–C2 angle. The combination of these two effects pushes the H3 atom farther away from the center of the molecule.

We next turn to the excited vibrational state obtained when the O–H stretch normal mode has acquired one quantum of vibrational energy. Figure 3 shows for this eigenstate the density difference between the anharmonic and the harmonic nuclear densities. The actual excited harmonic O–H stretch nuclear density is displayed in Supplementary Figs. 6b1–4, while Supplementary Figs. 6d1–4 shows the actual excited anharmonic

O–H stretch state nuclear density. Also in this case, anharmonic effects are distributed all over the molecule. In addition, we observe a general elongation of bond lengths by comparing the bond-length distributions obtained from the harmonic and the anharmonic excited state wavefunctions, especially for the CH$_2$ − NH$_3^+$ part of the molecule (see Fig. 3b, e, f). Once more, looking at the complete nuclear density picture in Fig. 3a, one can deduce that these elongations balance the density distortion due to anharmonicity along the O–H stretching direction. Remarkably, an interesting lobe pattern appears along the O1–H6 stretching direction, as can be rationalized with the inspection of bond-length distributions in Fig. 3b. Here, the bond length analysis confirms the global picture of anharmonicity obtained by visual inspection of density differences in Fig. 3a. The unidimensional plot reported in Fig. 3b compares the harmonic distribution (black continuous line) with the semiclassical anharmonic one, which presents the average shifted towards longer bond distances and hence it is represented with the green continuous line. In the harmonic picture, a node can be observed along the O–H stretch direction. This means that the vibrational excitation can be effectively approximated as the excitation of a harmonic one-dimensional oscillator whose potential varies along this direction. This agrees with the standard harmonic normal-mode picture which shows the major displacement along the O–H stretch

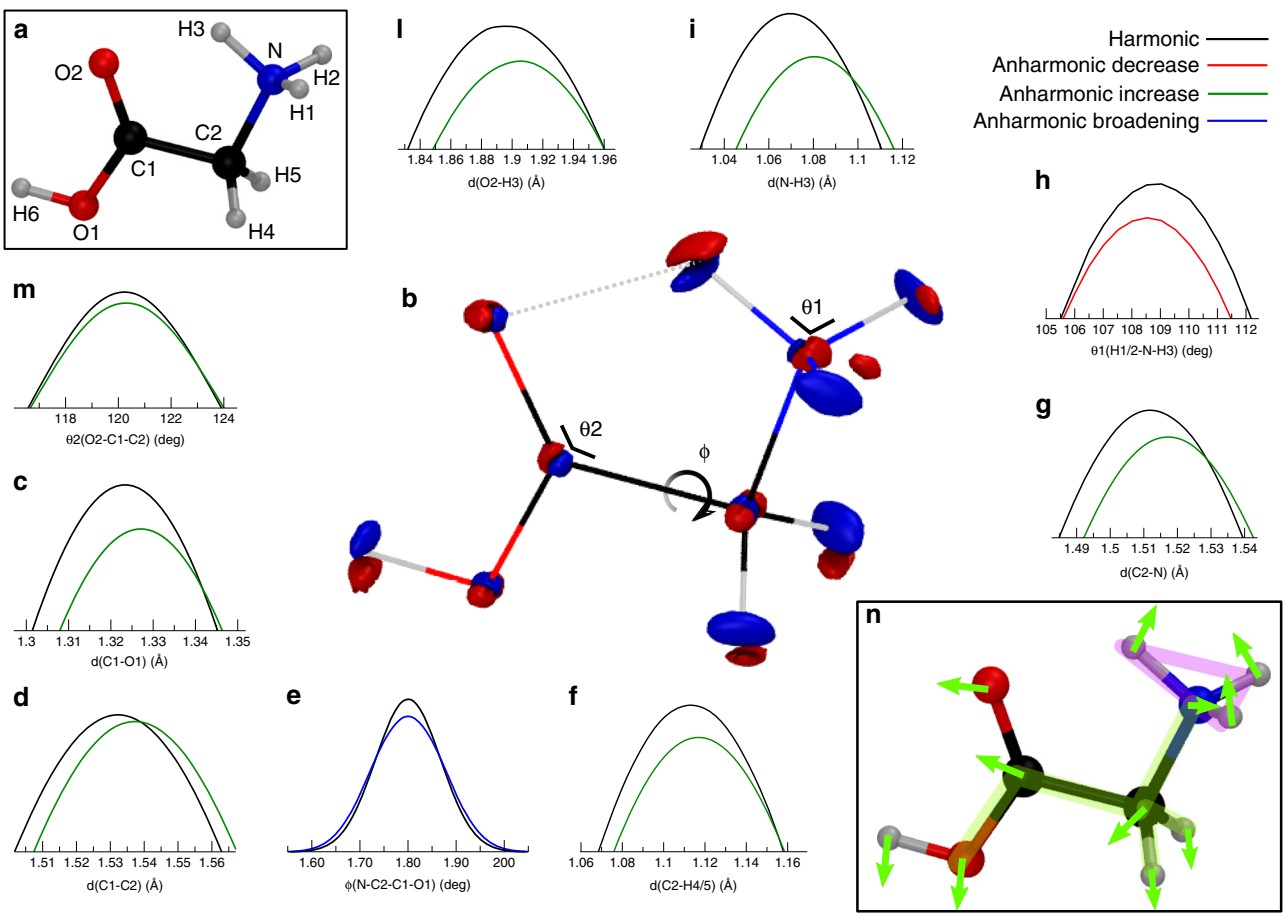

**Fig. 2 Comparison of harmonic and anharmonic nuclear densities for the vibrational ground state. a** The $C_s$ symmetry equilibrium geometry of GlyH$^+$ at DFT-B3LYP/aug-cc-pVDZ level of theory with the atomic labeling that we adopt as reference throughout the paper. **b** Two isosurfaces of the difference between the anharmonic and the corresponding harmonic nuclear densities for the vibrational ground state. Red indicates positive contributions, where the density concentrates due to anharmonicity, while blue stands for the negative contributions, where the density is depleted. The density isosurfaces are respectively set to 0.15 a.u. and −0.15 a.u. In addition, we report the comparison of the maxima of relevant geometry parameter distributions. **c, d, f, g, i, l** Bond lengths distributions. **h, m** bond angles distributions. **e** Dihedral distribution. The black curves are obtained from the harmonic approximation while the colored ones are calculated from the full anharmonic wavefunction. Green curves present the maximum shifted towards longer bond lengths or wider angles (Anharmonic increase), on the contrary red indicates contraction (Anharmonic decrease). Blue stands for broadening of the anharmonic distribution with respect to the harmonic one, without maximum shift (Anharmonic broadening). **n** Pictorial representation of the overall effect of anharmonicity and couplings on the quantum nuclear density which leads to specific nuclear density redistribution (indicated by the green arrows) giving two distinctive structural effects, the backbone elongation (highlighted in green) and the closing of NH$_3^+$ umbrella (highlighted in magenta). These are deduced from the consideration of geometrical parameters distributions reported in **c–i**, **m** and in the Supplementary Figs. 1, 2, 3.

direction in Cartesian coordinates, as reported in Supplementary Table 3. Instead, after inclusion of anharmonicity, we can barely detect a reminiscence of the node, and the density concentrates closer to the equilibrium position, as can be also seen in Fig. 3c, where the difference between the anharmonic and harmonic distributions is reported. In the anharmonic picture, the O–H stretch excitation can therefore not be regarded as the excitation of a single bond oscillation, and it is not sufficient, for instance, to consider a separable Morse potential along the O–H bond to model this vibration. We stress that the anharmonic distribution of Fig. 3b, which was obtained with the semiclassical approximation of the wavefunction, has a quantum nature even if the calculation is based on classical information. This is evident when we build a histogram by binning instantaneous bond lengths along the quasi-classical trajectory employed for the semiclassical calculation (i.e., the constant energy classical trajectory with excited O–H stretch kinetic energy in harmonic approximation, as described in the Supplementary Methods). The quasi-classical distribution is peaked at the turning points. On the contrary, the

semiclassical distribution clearly shows quantum features, since it extends beyond the classical turning points located at the extremes of the histogram, and it displays higher probability in the middle.

**Normal mode vs. quantum anharmonic O–H stretch excitation.** Finally, in Fig. 4 we propose a 3D real-space representation of the anharmonic motion which the molecule undergoes when it is excited by IR radiation, using the observation of nuclear density depletion and accumulation, as an improvement over the classical harmonic normal-mode displacements. Specifically, we calculate the difference between the anharmonic nuclear density of the O–H stretch excited vibrational state and the ground-state one (Fig. 4b, d). We also compare it to the corresponding quantum harmonic estimate, reported in Fig. 4a, c.

Considering that the excitation linked to the vibrational spectra signal at 3553 cm$^{-1}$ in the experimental IR spectra is interpreted as the O–H stretch, we first concentrate on the nuclear density deformation by changing the vibrational state from the ground to

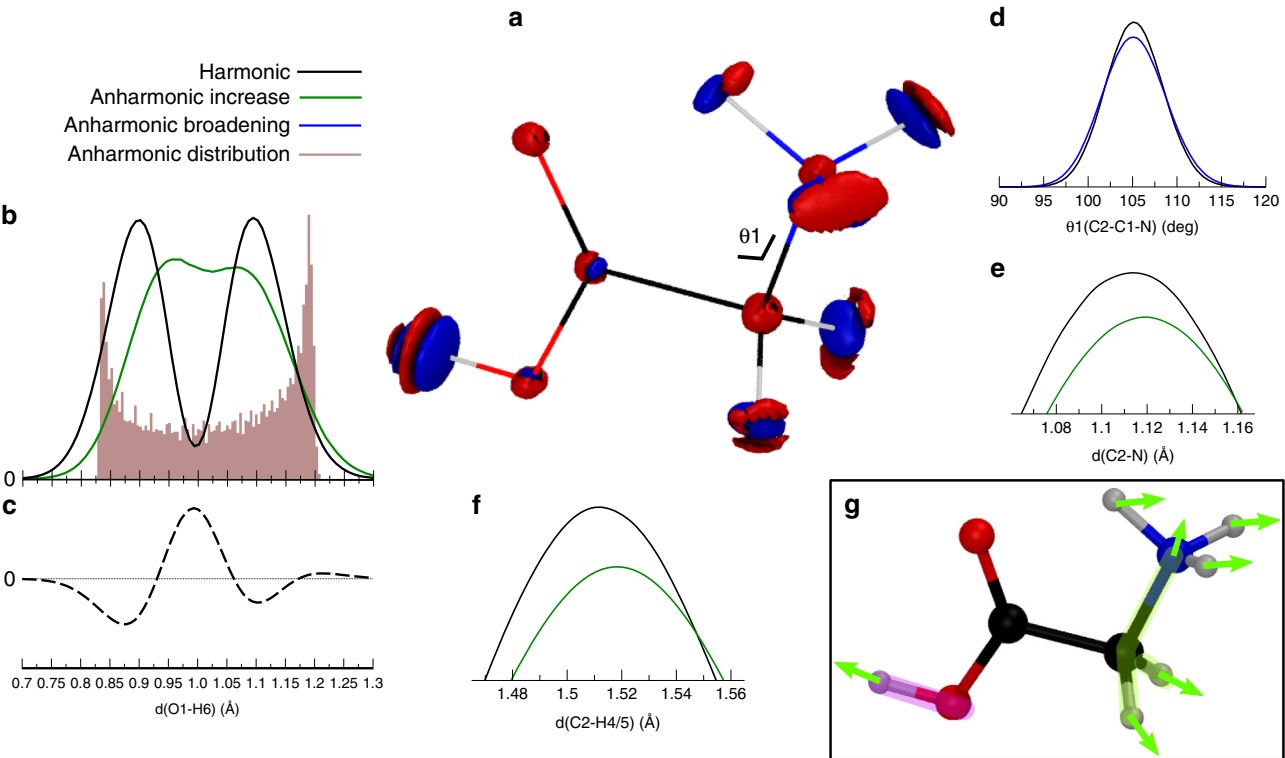

**Fig. 3 Comparison of harmonic and anharmonic nuclear densities for the O–H stretch vibrational excited state. a** The same as in Fig. 2b but for the O–H stretch vibrational excited state. **b–f** Relevant geometry parameter distributions with the same color code as in Fig. 2. The solid black curves are obtained from the harmonic approximation of the wavefunction, while the colored ones are calculated from the full anharmonic wavefunction. Green curves present the maximum shifted towards longer bond lengths (Anharmonic increase), while the blue one stands for broadening of the anharmonic distribution with respect to the harmonic one and without maximum shift (Anharmonic broadening). Specifically, **b** Bond-length distributions along the radial O1–H6 stretch distance. The filled area is the histogram of the classical distribution obtained from the quasi-classical trajectory used to generate the semiclassical wavefunction. The harmonic normal mode distribution is small but not equal to zero at equilibrium distance because the variation of O–H stretch normal mode does involve, even if minimally, also other atom displacements. **c** Difference between anharmonic and harmonic curves of **b**. The statistical error bars for the distributions are smaller than the line width. **d** C2–C1–N angle distribution. **f**, **e** enlargement of the distribution peaks for the C2–H4/5 and C2–N bond distances. **g** Summary of the overall effect of anharmonicity and couplings on the quantum nuclear density which leads to specific nuclear density redistribution (indicated by the green arrows) giving effects which counterbalance each other in two separate regions of the molecule, the hydroxyl (highlighted in magenta) and the $CH_2NH_3^+$ (highlighted in green). These are deduced from the consideration of geometrical parameters distributions reported in **b**, **f**, **e**, **d**, and in the Supplementary Figs. 4, 5.

the excited one. We start with an analysis focused on the hydroxyl group of the molecule (Fig. 4a, b). The harmonic case reported in Fig. 4a provides lobes with perpendicular nodal planes with respect to the O1–H6 bond. There is a minor density deformation contribution on the O6 nucleus, while the bigger one is on the lighter H6. Turning to the anharmonic picture shown in Fig. 4b, we find a similar pattern but with the lobes that have no longer parallel nodal planes. The lobes are in this case distributed along a curved line. This is an effect of anharmonicity and vibrational couplings which leads to a less symmetric oscillation of the O–H bond.

By zooming out to the whole molecule structure in Fig. 4d, we can appreciate how anharmonicity calls into play all the other nuclei. The difference between the harmonic picture of Fig. 4c is striking. The harmonic picture in panel c is compliant with the chemical intuition and with the classical normal-mode displacement picture. The excitation of the O–H stretch normal mode results in a very localized density deformation, which causes a depletion and a consequent increment of density exclusively for the hydroxyl functional group, while in the other parts of the molecule the ground state and the excited state densities cancels perfectly upon subtraction. Instead, the quantum mechanical anharmonic picture provided by the semiclassical vibrational

densities clearly indicates that all the nuclei are involved in the excitation, even if the anharmonic density difference exhibits (with distortions) the typical lobe pattern of the O1–H6 stretch excitation. To give a quantitative assessment of the involvement of each nucleus in the excitation in the harmonic and anharmonic case we can also refer to the density standard deviation position differences (see Supplementary Discussion) shown in Supplementary Tables 4, 5. These data confirm what is clearly visible from the nuclear density differences in Fig. 4c, d, i.e., that many more atoms are involved in the O–H stretch excitation when the anharmonic density is employed. More importantly, the anharmonic nuclear density picture highlights the relevant coupling of the hydroxyl group with the far protonated amino group, which is significantly involved in the O–H stretch vibrational excitation. This is proved by the values of the harmonic basis set functions reported in Supplementary Table 1, where the contribution of each normal mode to the spectroscopic signal at 3553 cm$^{-1}$ is quantified. The vibrational coupling is therefore non-local in this case, i.e., it is not confined to the nearest neighbor atoms as shown in Fig. 4e for the harmonic approximation. In particular, the anharmonic vibration of the hydroxyl functional group triggers the vibration of the $CH_2NH_3^+$ groups, as pictorially summarized by Fig. 4f.

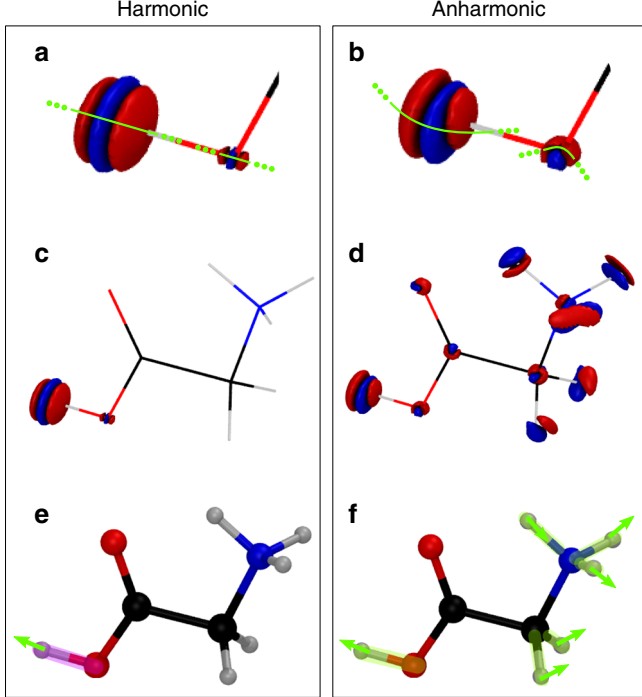

|Harmonic|Anharmonic|

**Fig. 4 Nuclear density difference plots for the excited O-H stretch vibrational eigenstate of the GlyH$^+$. a** Difference between the harmonic density of the excited eigenstate and the corresponding ground-state one on the hydroxyl group. **b** The same but for the anharmonic case. **c, d** The same differences but for the entire molecular structure. Red indicates positive contributions, while blue stands for the negative contributions. All isodensities are set to 0.15a.u. and −0.15a.u. **e, f** Pictorial representation of different interpretations of the vibrational excitation as derived by the harmonic picture (**e**) and the anharmonic one (**f**). The green arrows indicate the most significant shifts of the nuclear densities, while the magenta and green halos highlight the molecular regions most affected by the geometrical distortions caused by excitation respectively in the harmonic and anharmonic case. These are deduced from the consideration of geometrical parameters distributions reported in the Supplementary Figs. 1–5.

## Discussion

In conclusion, the present method allows for the calculation of high dimensional quantum eigenfunctions of vibrational ground and excited states for molecules of moderate size, and provides an immediately informative real space nuclear density representation of molecular vibrations, overcoming the major limitations of the normal-mode approach and going beyond the quantum harmonic picture[53].

The analysis of the quantum nuclear densities for a biologically important and experimentally significant molecule, GlyH$^+$, shows that anharmonic contributions, that can be assessed thanks to our semiclassical methodology, have a significant impact on the interpretation of relevant vibrational excitations. Overall, our anharmonic quantum simulation provides a flexible molecule picture with a greater adaptability to the various binding situations than the harmonic approximation. As an example, we calculate the eigenfunction corresponding to the free O–H stretch energy peak and find that other modes, in addition to the O–H stretch one, contribute to this spectroscopic signal. More specifically, the amino group plays a significant synergistic role in the O–H stretch excitation. Even if the mixing of normal modes is the well known consequence of the introduction of couplings, the determination of the amount of mixing as resulting from our calculations is not trivial, also given the position of the two

functional groups in the molecule, and the normal mode picture that we are accustomed to. Instead, our semiclassical method provides a quantum mechanical anharmonic picture of molecular vibrations which reveals quantitatively that more than one mode can be responsible for a given fundamental excitation and how much the modes are mixing. When we consider the complete potential, which includes all the anharmonicities without any other approximation apart from the level of electronic structure theory adopted, our calculations quantitatively characterize the involvement of other functional groups, also in the case of spectral features which are usually assigned to very localized normal modes, such as the O–H stretch one of the GlyH$^+$.

Eventually, this method allows a more accurate, physically sound assignment of fundamental and overtone vibrational absorption bands. This approach should stimulate a better rationalization of the experimental results, providing a reliable tool to gauge the extent and importance of couplings for a comprehensive understanding of anharmonic vibrational behavior in molecules.

## Methods

**Computational details**. The semiclassical calculation of GlyH$^+$ eigenfunctions is performed using on-the-fly classical trajectories at the DFT-B3LYP level of theory using the aug-cc-pVDZ basis set with the NWChem package[70]. We employ one trajectory for the ground state and another one for the O–H stretch excited state. For each one, the Hessian matrix is calculated at each time-step. For our DFT simulations the timing is: 16 h for the trajectory evolution and 552 h for the Hessians employing the NWChem code on 20 CPUs with clock frequency of 2.6 GHz. We also point out that the calculation of the Hessian along the trajectory is embarrassingly parallel, and the scaling with the number of cores employed is linear. Each GlyH$^+$ eigenfunction has been symmetrized so that it belongs to one of the $C_s$ irreducible representations and is given by a combination of 12,799 harmonic functions, which are obtained from considering all possible single and simultaneous excitations of two modes with maximum quantum number $n_\alpha = 6$. However, only the coefficients greater than $10^{-3}$ are important for the eigenfunction shape. The excited eigenfunction is orthonormalized via Gram-Schmidt with respect to the ground state one. The 3D histogram for the one-nucleus density is composed of cubes whose edge is 0.049 Å. We represent these histograms in the standard cube file format, which can be read by any visualization software which supports it. In particular, we used the VMD software to produce all the nuclear density pictures in the present work[71]. Instead, the density space resolution of the bond-length distributions in Fig. 3b is equal to 0.008 Å, the one for angle and dihedral distributions is 0.45 degrees. The evaluation of wavefunction coefficients and the calculation of the densities takes few minutes on a personal computer and it does not necessarily require many cores or large memory.

## Data availability

Any data generated and analyzed for this study that are not included in this Article and its Supplementary Information are available from the authors upon request.

## Code availability

The computer code and the data files used to produce all the figures presented in this study are available as Supplementary Code (zip file). The code is released under the GNU General Public Licence v3.

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

## Acknowledgements

The authors thank Prof. E. Garand for providing the experimental spectrum of protonated glycine. The authors thank Dr. F. Gabas for contribution in the early stages of this work, Prof. L. Lo Presti for discussion, and Prof. A. Gavezzotti for a critical reading of the manuscript. The authors acknowledge financial support from the European Research Council (ERC) under the European Union's Horizon 2020 research and innovation programme [Grant Agreement No. (647107)—SEMICOMPLEX—ERC-2014-CoG] and from the Italian Ministry of Education, University, and Research (MIUR) (FARE programme R16KN7XBRB project QURE). Part of the needed CPU time was provided by CINECA (Italian Supercomputing Center) under ISCRAB project QUASP, ISCRAC project MCSCMD, and ISCRAC project heavyTUN.

## Author contributions

C.A. run the simulations, analyzed the data, produced the figures and substantially contributed to the draft. M.M. conceived the idea and the computational method for representing molecular excitations through nuclear densities, and run preliminary simulations. G.B. contributed in the optimization of the software and the analysis of the data. M.C. supervised the work, interpreted the results, and wrote the draft. All authors discussed the results and commented on the manuscript.

## Competing interests

The authors declare no competing interests.
