## [Peer Review File · Nature Communications]

Reviewers' comments:

Reviewer #1 (Remarks to the Author):

In this interesting and well-written manuscript, the authors describe a rigorous procedure based on semiclassical dynamics to visualize nuclear probability densities. They use the procedure to illustrate clearly the difference between the harmonic and anharmonic vibrational wavefunctions. As their main result, the authors show that the anharmonic wavefunction with a single excitation in the O-H stretch mode, unlike its harmonic counterpart, involves multiple atoms, even those that are far away from the OH functional group. In particular, the method overcomes the major drawbacks of classical analysis (which ignores ZPE) and quantum analysis within the harmonic picture (which ignores anharmonicities). It is remarkable that in Fig. 2b, the density is captured even in the classically forbidden region. Although the effects of anharmonicity have been studied in many ways in the last few decades, the simple picture provided by the present method is most certainly worth attention and I support its publication.

I have several questions and suggestions:

- 1) Even in the harmonic analysis, the normal mode dominated by the OH stretch is not perfectly isolated from the motion of other atoms (because it is a normal, and not local mode, after all). In Fig. 4, it seems perfectly isolated (as far as my eyes can distinguish). Could the authors list numerically the relative displacements of all 11 atoms in the OH stretch normal mode within harmonic approximation, and within the anharmonic model (to fix the magnitude, consider the classical turning point at the ZPE) ? E.g., in a table in the supplementary information.
- 2) How is the "OH stretch" defined in the anharmonic model? To be complete, the authors should provide a precise definition that they used even for the harmonic model.
- 3) The authors present nuclear density differences in Figs. 2, 3, and 4 (between harmonic and anharmonic in the first two figures, between the ground and excited state in Fig. 4), but the actual nuclear densities are not shown. Could the authors show these densities analogously to Figs. 2b, 3a, 4a, and 4b.? It would be nice to see how important the change in the density is compared to the magnitude of the density.
- 4) In electronic structure theory, when comparing ground and excited states, one can either plot the difference density (such as those shown in Fig. 4) or the transition density. Could this method be used to compute and visualize the transition density between the vibrational states?
- 5) Since the list of references is already quite large, the authors could mention other works that visualize nuclear densities beyond the harmonic approximation.

Otherwise, I have several minor comments:

1. Are the densities plotted using a readily available software for visualizing, e.g., molecular orbitals or electronic densities? If yes, please mention it in Methods or in the SI. Otherwise, would it be possible, in general, to prepare the file containing nuclear density to be compatible with such softwares (e.g. GaussView, VMD, Avogadro, Molden, ...)?
2. On page 5, the sentence "Therefore, differently from classical normal mode displacements,..." is obscure. For me, quantum and classical normal mode coordinates are the same coordinates, Depending on whether one includes rotations and translations, either $3N$ or $3N-6$ ($3N-5$) in total. Could the authors expand on what difference they have in mind?
3. On page 8, in Fig. 1, I am not sure if the interruption of the x-axis scale is needed. It seems to me that only $\sim 10\%$ of the energy range is removed. If more space is saved, the authors should clearly indicate the range to the left of the interruption. (The problem is aggravated because the tick $20,000 \text{ cm}^{-1}$ is right under the interruption sign so it is not clear if it corresponds to the scale on the left or on the right.)
4. On page 9, the sentence "The value of ZPE energy ... is well above ...which is a huge amount of energy in comparison with the harmonic vibrational energy level spacing." is stated at the beginning of the page, but its significance is only explained in the first sentence of the next section (still on page 9) mentioning that the ZPE is overlooked in classical approaches. This is a great point but should be clarified immediately.
5. The title of Fig. 2 "...ZPE nuclear densities" should perhaps be changed to something clearer, e.g., in analogy to Fig. 1 to "...for the ground state nuclear densities"
6. In Fig. 3 the legend "Anharmonic increase" should be just "Anharmonic", right? The increase is only shown in panel c.

Reviewer #2 (Remarks to the Author):

The manuscript by Aieta et al. discusses a computational investigation on geometry and vibrational modes of protonated Glycine obtained through a quantum description of the nuclei that include anharmonic effects. The average geometry of the molecule is found to be slightly elongated with respect to classical results, and the vibrational OH stretching acquires (due to anharmonicity) components on other normal mode as well.

The computational achievement (based on a clever scheme developed previously by M. Ceotto's group) is certainly remarkable. What concerns me most is the value of the new insights obtained from such challenging calculations. It is stated that the presented results contradict the "group frequency hypothesis". I do not think this is the case: here it is shown that if anharmonicity is taken into account, the OH stretching has some degree of mixing with other modes (mostly NH). But this is barely surprising, one of the expected effects of anharmonicity is to mix modes that are independent at the harmonic level (and mixing modes that are the closest in energy is the most natural occurrence). Already at textbook level the concept of group frequency is understood to be a useful but oversimplified picture of reality, the present work confirms this view.

In terms of quantify the extent of the effect, pictures such as fig. 3a may be misleading: they give the impression that OH stretching and other modes are mixed with comparable weights, but these are only the anharmonic nuclear densities on top of the harmonic ones. If the total (harmonic+anharmonic) density would be plotted, I expect a less impressive mixing, with the isosurfaces on NH rather smaller than those on OH.

Concerning the ground state geometry modifications, again the results are quantitatively non trivial, but qualitatively expected: the asymmetry in bonding potentials, such as in Morse's, leads to change in the average bond length, even at the level of vibrational ground state.

There are also some specific issues:

1. Computational procedures such as those in ref. 59 for glycine have been used to provide accurate molecular geometries. It is unclear to me how the anharmonic corrections found here should be integrated in such schemes: are they additional corrections to be accounted for in such protocols?
2. From the supporting material, the procedure seems to provide also the coefficients of mixing of the various normal modes into the vibrational eigenfunction: could these be used to quantify the mixing of the OH stretching with the other modes?
3. Fig. 3b: The harmonic density distribution of the first vibrational excited state of the OH stretching has no node. Why is that so?
4. Fig. 2m: the pictorial representation using arrows of the anharmonic density plotted in fig. 2b is certainly useful. However I cannot understand why arrows on H4, H5 and H6 are missing, since there is anharmonic density also on these atoms.

In summary: I find the theoretical and computational achievement remarkable, the idea of representing the geometry and vibrational modes by real space nuclear densities effective and the new quantitative insights useful. Overall, all these points might warrant the broad interest expected for a Nature Communications paper. On the other hand, I think that the article in its present form is making too strong and unjustified claims, for the reasons I discussed above.

Reviewer #3 (Remarks to the Author):

The paper reminds me a set of old studies on X-H stretchings in polyatomic molecules, which could be described starting from either normal-modes or local-modes. The former are harmonic oscillators delocalized on the whole molecule, perturbed and coupled by quartic anharmonic interactions. The latter are anharmonic oscillators localized on individual X-H bonds with harmonic interbond coupling. It was discovered that two descriptions are completely equivalent and that their two Hamiltonians yields identical eigenvalues if their parameters obey proper relations (called x-K relations).

Exactly as in the present paper, it was noticed that when the energy is increased it tends to concentrates on individual bonds. I don't remember anymore the literature on this field, but it can be found by searching for "local modes and x-K relations". It could be useful to the authors.

Despite the fact that it is a nice and well written paper, I don't think this work is suitable for NatureComm. After citing the aforementioned old works, this paper can be submitted to a more specialized journal.

**REPLY TO REVIEWERS' COMMENTS: Anharmonic quantum nuclear densities
from full dimensional vibrational eigenfunctions: an unexpected picture for
protonated glycine vibrations**

Chiara Aieta, Marco Micciarelli, Gianluca Bertaina, and Michele Ceotto
Dipartimento di Chimica, Università degli Studi di Milano, via C. Golgi 19, 20133 Milano, Italy

In this document we report a point by point reply to Reviewers' comments which are highlighted in blue color.

I. REPLY TO REVIEWER'S #1 REMARKS

Reviewer #1: In this interesting and well-written manuscript, the authors describe a rigorous procedure based on semiclassical dynamics to visualize nuclear probability densities. They use the procedure to illustrate clearly the difference between the harmonic and anharmonic vibrational wavefunctions. As their main result, the authors show that the anharmonic wavefunction with a single excitation in the O-H stretch mode, unlike its harmonic counterpart, involves multiple atoms, even those that are far away from the OH functional group. In particular, the method overcomes the major drawbacks of classical analysis (which ignores ZPE) and quantum analysis within the harmonic picture (which ignores anharmonicities). It is remarkable that in Fig. 2b, the density is captured even in the classically forbidden region. Although the effects of anharmonicity have been studied in many ways in the last few decades, the simple picture provided by the present method is most certainly worth attention and I support its publication.

Authors: We thank the Reviewer for their comments, the constructive revision of our manuscript and for highlighting that the simple picture provided by our semiclassical approach can be valuable for studying the effects of anharmonicity.

Reviewer #1: I have several questions and suggestions: 1) Even in the harmonic analysis, the normal mode dominated by the OH stretch is not perfectly isolated from the motion of other atoms (because it is a normal, and not local mode, after all). In Fig. 4, it seems perfectly isolated (as far as my eyes can distinguish). Could the authors list numerically the relative displacements of all 11 atoms in the OH stretch normal mode within harmonic approximation, and within the anharmonic model (to fix the magnitude, consider the classical turning point at the ZPE) ? E.g., in a table in the supplementary information.

Authors: We thank the Reviewer for pointing out this issue. Their suggestion will surely help the reader for a better comprehension of the manuscript.

The Referee is right regarding the harmonic atomic displacement of the “O-H stretch” mode. This normal mode does not involve only these two atoms. Following the Reviewer’s suggestion, we consider the maximum elongation (turning point) geometry of the O-H stretch normal mode with a total energy equal to its harmonic ZPE. In Table RI we report the atomic Cartesian displacements (x, y, z components together with the magnitude $|\mathbf{d}|$) at the turning point with respect to the equilibrium geometry.

Table RI. Atomic displacements in Cartesian coordinates x, y, z for the maximum elongation (turning point) of the O-H stretch normal mode motion. The energy is set at the ZPE value of the mode. $|\mathbf{d}|$ is the displacement intensity, i.e. the squared root of the sum of the square of each component. Color code is red for the larger displacements, orange for the intermediate ones, and yellow for the small ones. The other displacements are negligible. The nuclei labels are the same as in Fig.2a of the MS.

Harmonic Displacements (\AA)				
Nuclei	x	y	z	$ \mathbf{d} $
N	-0.00004	-0.00002	0.00000	0.00005
C2	-0.00004	-0.00005	0.00000	0.00006
H4	-0.00007	0.00017	-0.00015	0.00024
H5	-0.00007	0.00017	0.00015	0.00024
C1	0.00019	0.00005	0.00000	0.00019
O2	-0.00006	0.00009	0.00000	0.00011
O1	0.00521	0.00407	0.00000	0.00661
H3	0.00025	0.00028	0.00000	0.00037
H1	0.00013	0.00001	-0.00021	0.00024
H2	0.00013	0.00002	0.00021	0.00025
H6	-0.07225	-0.05759	-0.00005	0.09240

In addition, Fig.R1 displays the geometry changes obtained after applying to the equilibrium geometry the displacements reported in the table above together with the reference frame (x red axes, y green axes and blue z axes).

Figure R1. O-H stretch maximum elongation (turning point) geometry (transparent representation) at ZPE value compared to the equilibrium geometry (filled representation). Reference setting frame is x for the red axes, y for the green axes and blue for the z axes. C atoms are in green, N in blue, O in red, and H in gray.

These results show that within the normal mode framework the O-H stretch is a fairly local movement which involves mainly the hydroxyl Hydrogen (H6) and Oxygen (O1). However, there are some minor displacements also for the other nuclei. The Hydrogen nuclei displacements are about ten times larger with respect to Nitrogen and Carbon ones because of the lighter mass. Thus, as the Reviewer pointed out, the normal mode is not completely local. However, we can safely assert that the normal mode stretching is mainly confined to the O-H group geometry changes. The nuclear density difference in Fig.4c in the MS highlights precisely these major components. We add both the normal mode displacement table and Fig.R1 in the Supporting Information (SI) and cite them in the revised MS on page 10 and 15.

The issue remains about how to estimate such type of displacements in the anharmonic motion case. Unfortunately, as the Reviewer noted in the question following below, there is no anharmonic equivalent to the harmonic modes to directly compare with. In other words, although we can associate a precise classical trajectory and a precise turning point geometry to the harmonic normal mode motion, this is not possible in the case of the anharmonic motion, where the coupling does not allow to distinguish each mode turning point separately. However, harmonic and anharmonic approaches can be compared on the same ground by using the nuclear density, as shown in the MS. To compare the harmonic and anharmonic motion and in a simplified but still quantitative way, we calculate the standard deviation of each nucleus position for the ground and the excited O-H stretch states, using the corresponding harmonic and anharmonic densities. Being the one-particle probability densities expressed in Cartesian coordinates, their standard deviations are vectors with three coordinates

$$\sigma_{\mathbf{R}_i} = (\sigma_{x_i}, \sigma_{y_i}, \sigma_{z_i}) \quad (1)$$

defined according to the usual standard deviation formula

$$\sigma_{\mathbf{R}_i}^2 = \overline{\mathbf{R}_i^2} - \overline{\mathbf{R}_i}^2, \quad (2)$$

where the i index counts the nuclei in the molecule. The averages in Eq.2 are calculated for a given vibrational state, using the same sampling geometries employed for the densities calculation. As anticipated, we calculate them for the ground state and for the O-H stretch excited state density distributions. A given atom is deemed to not take part to the O-H excitation when the change of its standard deviation is negligible. Tables RII and RIII report in the last four columns the difference of the three standard deviation components of the excited O-H stretch state with respect to the ground state for all nuclei, both assuming a harmonic (Tab.RII) or anharmonic (Tab.RIII) quantum approach. When the standard deviation variation is positive, nuclei are more delocalized in the excited state with respect to the ground state, while negative values are for nuclei which are more confined during the excited state motion with respect to the ground state ones.

Table RII. Harmonic atomic displacement standard deviations in Cartesian coordinates (units are in Ångström). First column indicates the atom according to the labeling in the main text. Columns two, three, and four for the ground state, and five, six and seven for the O-H excited state density. Last four columns report the standard deviation differences between the excited and the ground ones. We highlight in bold the components $> 10^{-6}$ Å in absolute value, and the corresponding nuclei.

Nuclei	Harm-ZPE			Harm-27			Harm-27 - Harm-ZPE			
	σ_x	σ_y	σ_z	σ_x	σ_y	σ_z	$\Delta\sigma_x$	$\Delta\sigma_y$	$\Delta\sigma_z$	$ \Delta\sigma $
N	9.17342E-02	6.38789E-02	6.77726E-02	9.17343E-02	6.38788E-02	6.77723E-02	9E-08	-1E-07	-3E-07	3E-07
C2	7.09344E-02	7.91868E-02	6.77807E-02	7.09343E-02	7.91868E-02	6.77810E-02	-6E-08	4E-08	3E-07	3E-07
H4	7.10003E-02	9.01194E-02	5.25694E-02	7.10004E-02	9.01200E-02	5.25703E-02	8E-08	7E-07	9E-07	1E-06
H5	7.10029E-02	9.01121E-02	8.30739E-02	7.10030E-02	9.01116E-02	8.30744E-02	8E-08	-5E-07	5E-07	7E-07
C1	5.03531E-02	6.29595E-02	6.77754E-02	5.03533E-02	6.29596E-02	6.77753E-02	1E-07	1E-07	-1E-08	2E-07
O2	5.31665E-02	4.23752E-02	6.77742E-02	5.31665E-02	4.23751E-02	6.77744E-02	-5E-09	-3E-08	2E-07	2E-07
O1	3.07387E-02	7.42014E-02	6.77741E-02	3.07387E-02	7.42013E-02	6.77738E-02	5E-08	-2E-07	-3E-07	3E-07
H3	8.51134E-02	4.72236E-02	6.78386E-02	8.51130E-02	4.72235E-02	6.78389E-02	-4E-07	-1E-07	3E-07	5E-07
H1	1.01850E-01	6.60561E-02	5.36257E-02	1.01849E-01	6.60555E-02	5.36254E-02	-4E-07	-6E-07	-3E-07	8E-07
H2	1.01861E-01	6.60603E-02	8.19970E-02	1.01862E-01	6.60604E-02	8.19965E-02	3E-07	3E-08	-5E-07	6E-07
H6	1.79624E-02	6.35284E-02	6.78215E-02	1.80054E-02	6.35362E-02	6.78210E-02	4E-05	8E-06	-4E-07	4E-05

Table RIII. The same as Table RII but for the anharmonic density.

Nuclei	Anharm-ZPE			Anharm-27			Anharm-27 - Anharm-ZPE			
	σ_x	σ_y	σ_z	σ_x	σ_y	σ_z	$\Delta\sigma_x$	$\Delta\sigma_y$	$\Delta\sigma_z$	$ \Delta\sigma $
N	9.18939E-02	6.39329E-02	6.77734E-02	9.18277E-02	6.38735E-02	6.77734E-02	-7E-05	-6E-05	1E-08	9E-05
C2	7.09466E-02	7.92046E-02	6.77830E-02	7.09383E-02	7.92270E-02	6.77823E-02	-8E-06	2E-05	-7E-07	2E-05
H4	7.09279E-02	9.02778E-02	5.25719E-02	7.10377E-02	9.02226E-02	5.25612E-02	1E-04	-6E-05	-1E-05	1E-04
H5	7.09305E-02	9.02702E-02	8.30876E-02	7.10400E-02	9.02150E-02	8.31010E-02	1E-04	-6E-05	1E-05	1E-04
C1	5.03108E-02	6.28936E-02	6.77753E-02	5.02908E-02	6.29181E-02	6.77753E-02	-2E-05	2E-05	1E-08	3E-05
O2	5.30949E-02	4.22986E-02	6.77748E-02	5.31462E-02	4.23407E-02	6.77748E-02	5E-05	4E-05	-2E-08	7E-05
O1	3.06814E-02	7.42477E-02	6.77753E-02	3.07026E-02	7.42436E-02	6.77750E-02	2E-05	-4E-06	-3E-07	2E-05
H3	8.52621E-02	4.70354E-02	6.78338E-02	8.52897E-02	4.72181E-02	6.78404E-02	3E-05	2E-04	7E-06	2E-04
H1	1.02020E-01	6.60615E-02	5.37098E-02	1.02056E-01	6.60629E-02	5.35979E-02	4E-05	1E-06	-1E-04	1E-04
H2	1.02031E-01	6.60654E-02	8.19276E-02	1.02067E-01	6.60663E-02	8.20425E-02	4E-05	9E-07	1E-04	1E-04
H6	1.78328E-02	6.37068E-02	6.78191E-02	1.76824E-02	6.33426E-02	6.78221E-02	-2E-04	-4E-04	3E-06	4E-04

By comparing Table RII with Table RIII, we can see by highlighting in bold all displacements $> 10^{-6}$ Å that many more atoms are involved in the O-H stretch excitation when the anharmonic density is employed. However, the standard deviations in Tables RII and RIII can not be compared with the displacements in Table RI, since the maximum elongation is not directly comparable with the standard atomic amplitude deviation. We think that this information is very useful and helpful to better understand our results, and we decided to include them into the Supporting Information (SI) and briefly discuss them in the main text as shown in the tracked change MS on page 17.

We would also like to clarify the meaning of the arrows in Figs.4e and 4f, as well as in Figs.2m and 3g in the MS. These arrows are intended to help the reader to appreciate in a qualitative way what the significant nuclear density shifts for each nucleus are. In the same figures, the colored halos around the atoms are inferred by the analysis of all the length, angle and dihedral density distributions. They are meant to show in a pictorial way the main geometry distortions deduced by this information, both in the case of harmonic (Fig.4e of the MS) and anharmonic (Fig.4f of the MS) excitation, or due to the inclusion of anharmonicity in the potential of the ground and the O-H stretch

excited states (respectively Fig.2m and 3g of the MS). In a few words, our intention was not to use the arrows to indicate the normal mode displacements (in the case of Fig.4e of the MS) and the arrows extensions are not derived by any precise numerical assessment of geometry distortions. They just convey qualitative information. We updated the MS and clarify these issues in the caption of Figs. 2, 3, and 4.

Reviewer #1: 2) How is the “OH stretch” defined in the anharmonic model? To be complete, the authors should provide a precise definition that they used even for the harmonic model.

Authors: We start our reply from the harmonic model. As usual, the normal coordinates are obtained by diagonalizing the mass-scaled potential Hessian matrix \mathbf{H} at equilibrium

$$H_{i,j} = \frac{1}{\sqrt{m_i m_j}} \left(\frac{\partial^2 V}{\partial x_i \partial x_j} \right)_{eq} \quad (3)$$

The conversion matrix \mathbf{C} between the Cartesian coordinates \mathbf{x} and the normal coordinates \mathbf{Q} is such that $\mathbf{Q} = \mathbf{C}^T \mathbf{M}^{1/2} \mathbf{x}$, where \mathbf{M} is the diagonal matrix of the atomic masses and \mathbf{C} is the matrix whose columns are the eigenvectors of \mathbf{H} . For the GlyH⁺, the “O-H stretch” harmonic wavefunction is given by the direct product of 27 one-dimensional harmonic eigenfunctions, each one dependent on a different normal coordinate. The first 26 terms are ground state eigenfunctions and the 27th eigenfunction, whose normal-mode coordinate corresponds to the eigenvalue whose square root is the O-H stretch frequency value, is a first excited vibrational eigenfunction. The normal mode coordinate Q_{27} is given by a combination of mass weighted Cartesian coordinates with larger coefficients in correspondence of the coordinates of H1 and O6 atoms, as can be seen from the components of the conversion matrix elements \mathbf{C}^T along the 27th line reported in Table RIV. We now provide as a supplementary file the complete \mathbf{C} matrix and the equilibrium geometry used for all the calculations reported in the MS.

Table RIV. Conversion matrix elements from Cartesian coordinates to O-H stretch normal mode in atomic units. The larger coefficients are highlighted in bold.

	x	y	z
N	1.69E-03	8.18E-04	6.36E-06
C2	1.39E-03	1.87E-03	2.40E-06
H4	7.36E-04	-1.78E-03	1.58E-03
H5	7.45E-04	-1.78E-03	-1.58E-03
C1	-6.74E-03	-1.81E-03	-1.12E-05
O2	2.24E-03	-3.31E-03	3.02E-06
O1	-1.89E-01	-1.48E-01	-1.18E-04
H3	-2.62E-03	-2.92E-03	4.59E-06
H1	-1.39E-03	-1.57E-04	2.16E-03
H2	-1.40E-03	-1.62E-04	-2.18E-03
H6	7.59E-01	6.05E-01	4.86E-04

Now we reply to the main question, i.e. how is the O-H anharmonic stretch defined. We recall that our anharmonic wavefunction are given by a suitable linear combination of harmonic wavefunctions, i.e. $|e_n\rangle = \sum_{\mathbf{K}} C_{n,\mathbf{K}} |\phi_{\mathbf{K}}\rangle$. The expansion coefficients are calculated by collecting the power spectrum intensities at a specific energy, as specified in the SI, Section III. The anharmonic O-H stretch coefficient combination is obtained by looking at the value of each coefficient obtained from power spectra simulated with the Time-Average Semiclassical Initial Value Representation (TA SCIVR) by fixing the energy in correspondence of the O-H stretch fundamental peak, which is equal to 3,539cm⁻¹ above the ZPE, as described in Eq.19 of the SI. Motivated by the Referee’s question, we report in Table RV the nine larger expansion coefficients $C_{n,\mathbf{K}}$ for the anharmonic wavefunction of the O-H stretch. The largest coefficient in the Table is that one where $k_{27}=1$ and $k_i=0$, $i=1\dots 26$. This confirms that the correct assignment of the signal is an anharmonic O-H stretch excitation. The Table shows also the most important contributions to the wavefunction which derive from other 27 harmonic states. Therefore, we think the method we present can be useful also in the assignment of the semiclassical anharmonic spectra. In addition, this analysis confirms that the assignment derived by inspection of the nuclear densities with the method we developed is indeed effective. Specifically, in Fig.4 of the

Table RV. The first nine larger (in modulus) expansion coefficients for the anharmonic excited O-H stretch wavefunction. The most important contribution is given by the first row which corresponds to one quantum excitation of normal mode 27 (the O-H stretch one) and zero excitation for all the other normal modes.

\mathbf{K}	k_1	k_2	k_3	k_4	k_5	k_6	k_7	k_8	k_9	k_{10}	k_{11}	k_{12}	k_{13}	k_{14}	k_{15}	k_{16}	k_{17}	k_{18}	k_{19}	k_{20}	k_{21}	k_{22}	k_{23}	k_{24}	k_{25}	k_{26}	k_{27}	$C_{n=27,\mathbf{K}}$	
0	0	0	0	0	0	0	0	0	0	0	0	0	0	0	0	0	0	0	0	0	0	0	0	0	0	0	1	-6.09E-01	
1	0	0	0	0	0	0	0	0	0	0	0	0	0	0	0	0	0	0	0	0	0	0	0	0	0	0	1	0	3.53E-01
0	0	0	0	0	0	0	0	0	0	1	0	0	0	2	0	0	0	0	0	0	0	0	0	0	0	0	0	0	2.48E-01
0	0	0	0	0	0	0	2	0	0	0	0	0	0	0	0	0	0	0	0	0	1	0	0	0	0	0	0	0	-1.77E-01
0	0	0	1	0	0	0	0	0	0	0	0	0	0	0	0	0	0	0	0	0	0	0	1	0	0	0	0	0	-1.44E-01
0	0	0	0	0	0	0	0	0	0	0	0	0	0	0	0	0	0	0	0	0	0	0	0	0	0	0	2	1.29E-01	
0	0	0	0	0	0	0	1	0	0	0	0	0	0	2	0	0	0	0	0	0	0	0	0	0	0	0	0	0	-1.20E-01
0	0	0	1	0	0	0	0	0	0	0	0	0	0	0	0	0	0	0	0	0	0	1	0	0	0	0	0	0	-1.11E-01
0	0	1	0	0	0	0	0	0	0	0	0	0	0	0	0	0	0	0	0	0	0	0	0	0	1	0	0	1.02E-01	

MS one can appreciate the overall character of the anharmonic excitation from the shape of the lobes on the H6 and O1 and the changes of the O1 and H6 atom densities are more evident with respect to the other atoms, including the protonated amine umbrella ones.

We hope that these considerations clarify how we assign the O-H stretch excited state density and show how the method is useful for the assignment of the semiclassical anharmonic spectra as well. In a few words, the coefficients of the wavefunction and the visualization of the nuclear densities spot in a quantum and full dimensional way the character of the spectroscopic signal. We better explain our definition of the O-H stretch mode in the MS with the help of Table RV which is now added to the Supplementary Information. Eventually, we add the following phrases in the “**Result**” section:

“*In the harmonic case, the 27-dimensional vibrational Hamiltonian becomes separable in the normal-mode coordinates. Therefore, the harmonic wavefunctions are the direct product of 27 one-dimensional eigenfunctions, each one depending on a single normal-mode coordinate. The degree of excitation of each one-dimensional wavefunction defines the state. For instance, for the harmonic ground state, all the eigenfunctions in the product are the ground state solution of each separate one-dimensional Schrödinger equation with harmonic potential. Instead, the harmonic O-H stretch excited state is the same direct product as the ground state for the first 26 DOFs, but with the eigenfunction depending on the 27-th normal-mode coordinate in the first excited state solution. For the anharmonic case, we represent the wavefunction with a combination of harmonic wavefunctions, as defined in Eq.14 of the Supplementary Information. The combination coefficients are obtained with the semiclassical procedure described in the Supplementary Sections I and III. The character of the anharmonic wavefunction is determined by considering the relative contribution of the different harmonic basis functions. In particular, the ground state wavefunction has the largest coefficient on the harmonic ground state (see Table SI of the Supplementary Information), while the O-H stretch excited state has the largest coefficient on the harmonic basis function with one quantum of excitation on the 27th component which depends on the 27th normal mode and whose displacements correspond to the stretching of the Hydroxyl group (see Table SII of the Supplementary Information). However, in the latter case, we find other important contributions coming from other harmonic states that are mixing with the fundamental O-H stretch one. For example, mode 26 which is the asymmetric N-H stretching at fundamental frequency equal to $3,504\text{ cm}^{-1}$, mode 25, which is the symmetric N-H stretching at frequency $3,445\text{ cm}^{-1}$, and modes 23 and 22 which are the C-H symmetric and asymmetric stretching with harmonic frequencies equal to $3,116\text{ cm}^{-1}$ and $3,105\text{ cm}^{-1}$ respectively form combination states with low frequency modes that have large coefficients in the anharmonic wavefunction expansion. Also, there is a contribution from the overtone of the O-H stretching with two quanta of excitation.*”

Reviewer #1: 3) The authors present nuclear density differences in Figs. 2, 3, and 4 (between harmonic and anharmonic in the first two figures, between the ground and excited state in Fig. 4), but the actual nuclear densities are not shown. Could the authors show these densities analogously to Figs. 2b, 3a, 4a, and 4b.? It would be nice to see how important the change in the density is compared to the magnitude of the density.

Authors: We thank the Reviewer for this clever suggestion, which we promptly implement. First, we report in Table (RVI) below the maximum value for the density found for each state, which are the harmonic ground state (Harm-ZPE), the harmonic O-H stretch excited state (Harm-27), the anharmonic ground state (AnH-ZPE) and the

Table RVI. Maximum value of the nuclear density for the GlyH⁺ vibrational states considered in the MS.

State	Maximum value for the nuclear density (a.u.)
Harm-ZPE	72.522
Harm-27	72.533
AnH-ZPE	153.457
AnH-27	142.657

Figure R2. Plot of the nuclear density for different isosurface values. The first column identifies the vibrational eigenstate. Harm-ZPE stands for the harmonic approximated density for the ground eigenstate, while AnH-ZPE the corresponding anharmonic one. Harm-27 is the harmonic approximated density for the first excited vibrational eigenstate, and AnH-27 the corresponding anharmonic one.

anharmonic excited state (AnH-27). All the nuclear densities are normalized such that the integral of the total density gives the number of nuclei in the molecule, i.e. 11.

For each state we report in Fig.R2 four isosurfaces of the nuclear density plotted on the same scale. The heavier nuclei (Oxygen, Carbon and Nitrogen) show a very sharp density localized on each nucleus, while the Hydrogen ones have a much broader density and consequently smaller maxima. This is evident both at the harmonic and anharmonic level. It is interesting to see that in the harmonic case a node is found in the nuclear density on the Hydroxyl H in

the excited vibrational state (Fig.R2b4), while this is not the case when we look at the anharmonic nuclear density of the corresponding excited state (Fig.R2d4), because now the oscillation along the O-H bond is coupled with other normal mode degrees of freedom. The same conclusions have been reached in Fig.3b in the MS. We recall that the isosurface differences we show in the MS are set at 0.15 a.u. These plots and the caption have been added to the Supplementary Information and in the MS we refer to these plots when describing Fig.2 and Fig.3 in the revised MS on page 11 and page 13.

Reviewer #1: 4) In electronic structure theory, when comparing ground and excited states, one can either plot the difference density (such as those shown in Fig. 4) or the transition density. Could this method be used to compute and visualize the transition density between the vibrational states?

Authors: This is a very interesting suggestion and we have looked into this quantity. The time independent transition density is defined (see for instance for electronic transitions Yonghui Li, C.A. Ullrich *Chemical Physics* **391**, 157 (2011)) via the Time (**Independent**) Density Matrix (TDM) associated with an electronic transition between the ground state and the nth excited state, and it is equal to $T(\mathbf{R}) = \langle \Psi_n | \rho(\mathbf{R}) | \Psi_0 \rangle$. In the case of vibrational transitions, we can modify Eq.13 of the SI to obtain the following expression as

$$\rho_{n,m,\mathbf{R}_i}(\mathbf{R}) = \int d^{3N} \mathbf{Q} \langle e_m | \mathbf{Q} \rangle \langle \mathbf{Q} | e_n \rangle \delta(\mathbf{Q}^{RT}) \delta(\mathbf{R}_i(\mathbf{Q}) - \mathbf{R}) \quad (4)$$

which is the time independent transition density between two different vibrational states n and m . We can now introduce the bins in Cartesian space and the index function $I_{\mathbf{R}_i}^j(\mathbf{Q})$, as in Eq. 16 of the SI

$$\bar{\rho}_{n,m,\mathbf{R}_i}^j = \frac{1}{\Omega} \int d^{3N} \mathbf{Q} \langle e_m | \mathbf{Q} \rangle \langle \mathbf{Q} | e_n \rangle \delta(\mathbf{Q}^{RT}) I_{\mathbf{R}_i}^j(\mathbf{Q}) \quad (5)$$

where m and n vibrational eigenfunctions are again written as combinations of harmonic eigenfunctions $|e_n\rangle = \sum_{\mathbf{K}} C_{n,\mathbf{K}} |\phi_{\mathbf{K}}\rangle$ and $|e_m\rangle = \sum_{\mathbf{K}'} C_{m,\mathbf{K}'} |\phi_{\mathbf{K}'}\rangle$. Note that the basis set dimension can be different for the two states. By substituting these expansions, and by recalling Eq. 10 of the SI we obtain

$$\bar{\rho}_{n,m}^j = \frac{1}{\Omega} \int [d^{3N} \mathbf{Q} |G(\mathbf{Q}, \mathbf{\Gamma})|^2] \left(\sum_{\mathbf{K}} C_{n,\mathbf{K}} \bar{\phi}_{\mathbf{K}}(\mathbf{Q}) \right) \left(\sum_{\mathbf{K}'} C_{m,\mathbf{K}'} \bar{\phi}_{\mathbf{K}'}(\mathbf{Q}) \right) \delta(\mathbf{Q}^{RT}) I^j(\mathbf{Q}) \quad (6)$$

where we have employed the property that the coefficients of the expansion are real. Eventually, we get the final working formula for the calculation of the TDM between vibrational states

$$\bar{\rho}_{n,m}^j = \lim_{L \rightarrow \infty} \frac{1}{\Omega L} \sum_{l=1}^L \left(\sum_{\mathbf{K}} C_{n,\mathbf{K}} \bar{\phi}_{\mathbf{K}}(\mathbf{Q}_l) \right) \left(\sum_{\mathbf{K}'} C_{m,\mathbf{K}'} \bar{\phi}_{\mathbf{K}'}(\mathbf{Q}_l) \right) I^j(\mathbf{Q}_l) \quad (7)$$

where L independent molecular configurations \mathbf{Q}_l are sampled with the Box-Muller algorithm by a multivariate Gaussian distribution with null mean and variance equal to $(2\Gamma/\hbar)^{-1}$.

Unfortunately, we can not calculate the time-**dependent** Density Matrix TDM (see again Yonghui Li, C.A. Ullrich *Chemical Physics* 391, 157 (2011)) with the data that we have, because we do not have at the moment a way to calculate the time quantum evolution of the nuclear density. This calculation would involve the development of a method for the time evolution of the coefficient expansion for the vibrational states, a debugging process where analytical models are verified, and then a testing procedure against exact benchmark calculations. It is certainly an interesting project, that we will address as a continuation of this project.

Reviewer #1: 5) Since the list of references is already quite large, the authors could mention other works that visualize nuclear densities beyond the harmonic approximation.

Authors: We agree with the Reviewer that the main topic to be cited is about the visualization of nuclear densities. We change the phrase in the last part of the introduction on page 6 of the MS “*The Cartesian coordinate space representation allows the visualization of probability density isosurfaces in 3D in a similar fashion as routinely done for electron density.*[54,55]”

as

“*The Cartesian coordinate space representation allows the visualization of probability density isosurfaces in 3D, as for the nuclear ground state distributions obtained from Diffusion Monte Carlo calculations,[31,37,54-58], and in a similar fashion as routinely done for electron density.[59-60]*”

where we include a new group of citations ([31,37,54-58] in the revised MS new numbering) of other works where the nuclear density is visualized in the 3D space.

Reviewer #1: Otherwise, I have several minor comments:

1. Are the densities plotted using a readily available software for visualizing, e.g., molecular orbitals or electronic densities? If yes, please mention it in Methods or in the SI. Otherwise, would it be possible, in general, to prepare the file containing nuclear density to be compatible with such softwares (e.g. GaussView, VMD, Avogadro, Molden, ...)?

Authors: Our program outputs the density files in the standard cube file format. We attached a zip archive containing the program and the data needed for reproducing the figures of the manuscript, as requested by the policies of Nature Communications. In addition, our code can combine different cube files (i.e. linear combination of two cube files) to produce a new cube file containing, for instance, the density differences that we show in the MS. The cube file is a standard format that can be read by many visualization software. In our case, we used VMD to produce all the plots in the MS. As the Reviewer suggested, we added a note about the density file format and the visualization procedure in the “Methods” section of the MS.

Reviewer #1: 2. On page 5, the sentence “Therefore, differently from classical normal mode displacements, . . .” is obscure. For me, quantum and classical normal mode coordinates are the same coordinates, Depending on whether one includes rotations and translations, either $3N$ or $3N-6$ ($3N-5$) in total. Could the authors expand on what difference they have in mind?

Authors: We agree with the Reviewer that the phrase may sound obscure. When we wrote that sentence, we meant that positions and momenta are not uniquely defined for the normal mode vibrational quantum motion, differently from the classical trajectory that corresponds to the normal mode oscillation. Nevertheless, in quantum dynamics, vibrational normal modes are usually labeled as bending, stretching, wagging etc., and these definitions are given by considering the classical trajectories that correspond to the harmonic normal mode displacements. In a few words, even though one can define a normal mode coordinate system in a quantum treatment, one cannot identify any sort of trajectory that characterizes the normal modes. Also, considering that each atom is described by three degrees of freedom, the vibrational wavefunction is a high dimensional function even for small molecular systems and it can not be easily represented. Therefore, our 3D representation of nuclear densities provides a real space information originated from the wavefunction within a quantum mechanical formalism, that is without recurring to any sort of approximated classical picture, such as the trajectories derived from the normal mode displacements.

At the light of these considerations, we rephrase the original sentence

“Therefore, differently from classical normal-mode displacements, they are not directly interpreted as molecular motions represented in Cartesian space.”

as

“Therefore, differently from the classical picture which can be represented by classical trajectories, the normal-mode displacements are not directly interpreted as single point atom displacements in Cartesian space.”

Reviewer #1: 3. On page 8, in Fig. 1, I am not sure if the interruption of the x-axis scale is needed. It seems to me that only $\sim 10\%$ of the energy range is removed. If more space is saved, the authors should clearly indicate the range to the left of the interruption. (The problem is aggravated because the tick $20,000 \text{ cm}^{-1}$ is right under the interruption sign so it is not clear if it corresponds to the scale on the left or on the right.)

Authors: We used the interruption to neglect the portion of the spectrum between 0 cm^{-1} and $20,000 \text{ cm}^{-1}$. We choose to include the origin in this picture to show the bottom of the well represented in panel (b) and, in this way, to highlight the amount of ZPE for this system. We understand from the Reviewer’s comment that the present picture can be misleading and we have modified Fig.1 of the MS according to the Reviewer’s suggestion, by shifting the interruption sign to the left of $20,000 \text{ cm}^{-1}$ tick.

Reviewer #1: 4. On page 9, the sentence “The value of ZPE energy . . . is well above . . . which is a huge amount of energy in comparison with the harmonic vibrational energy level spacing.” is stated at the beginning of the page, but its significance is only explained in the first sentence of the next section (still on page 9) mentioning that the ZPE is overlooked in classical approaches. This is a great point but should be clarified immediately.

Authors: We agree with the Reviewer that we need to stress this point further and we modify the sentence

“The value of the ZPE energy from the global molecular minimum is well above $20,000$ wavenumbers, which is a huge amount of energy in comparison with the harmonic vibrational level spacing.”

as follows

“The value of the ZPE energy from the global molecular minimum is well above $20,000$ wavenumbers, which is a huge amount of energy in comparison with the harmonic vibrational level spacing. This quantum quantity cannot be

grasped by any classical simulation and it determines the physical behavior of the system as we will show in the next section. Also, here, we show the consequences of the harmonic approximation of this key quantity.”

Reviewer #1: 5. The title of Fig. 2 “...ZPE nuclear densities” should perhaps be changed to something clearer, e.g., in analogy to Fig. 1 to “...for the ground state nuclear densities”

Authors: Following the Reviewer’s suggestion we changed the title of Fig.2 in the MS from

“Comparison of harmonic and anharmonic ZPE nuclear densities.”

to

“Comparison of harmonic and anharmonic nuclear densities for the vibrational ground state.”

Reviewer #1: 6. In Fig. 3 the legend “Anharmonic increase” should be just “Anharmonic”, right? The increase is only shown in panel c.

Authors: In the legend of Fig.3 in the MS we use the words “increase” or “broadening” to indicate the main effect of anharmonicity on the nuclear density distributions, as we did for the legend in Fig.2 of the MS. Specifically, “anharmonic increase” means that the anharmonic distribution is shifted towards larger bond distances (or wider angle widths) with respect to the harmonic average, while “broadening” means that the anharmonic distribution is more spread than the corresponding harmonic one. To make this point clearer, we expand the caption of Fig.2 and Fig.3 of the MS.

We add the bold highlighted part in the caption of Fig. 2 as follows:

“Comparison of harmonic and anharmonic nuclear densities for the vibrational ground state. Panel a: The C_s symmetry equilibrium geometry of $GlyH^+$ at DFT-B3LYP/aug-cc-pVDZ level of theory with the atomic labeling that we adopt as reference throughout the paper. Panel b: Two isosurfaces of the difference between the anharmonic and the corresponding harmonic nuclear densities for the vibrational ground state. Red indicates positive contributions, where the density concentrates due to anharmonicity, while blue stands for the negative contributions, where the density is depleted. The density isosurfaces are respectively set to 0.15a.u. and -0.15a.u.. In addition, we report the comparison of the maxima of relevant geometry parameter distributions. Panels c, d, f, g, i, l: Bond lengths distributions. Panels h, m: bond angles distributions. Panel e: dihedral distribution. The black curves are obtained from the harmonic approximation while the colored ones are calculated from the full anharmonic wavefunction. Green curves present the maximum shifted towards longer bond lengths or wider angles (**Anharmonic increase**), on the contrary red indicates contraction (**Anharmonic decrease**). Blue stands for broadening of the anharmonic distribution with respect to the harmonic one, without maximum shift (**Anharmonic broadening**). Panel n: Pictorial representation of the overall effect of anharmonicity and couplings on the quantum nuclear density which leads to specific nuclear density redistribution (indicated by the green arrows) giving two distinctive structural effects, the backbone elongation (highlighted in green) and the closing of NH_3^+ umbrella (highlighted in magenta). These are deduced from the consideration of geometrical parameters distributions reported in panels c, d, e, f, g, h, i, m and the Section IV of the Supporting Information. These are deduced from the consideration of geometrical parameters distributions reported in panels b, f, e, d, and in the Section IV of the Supporting Information.”

We then modify the caption of Fig. 3 by adding the lines (in bold) as follows:

“Comparison of harmonic and anharmonic nuclear densities for the O-H stretch vibrational excited state. Panel a: the same as in Fig.2 but for the O-H stretch vibrational excited state. Panels b, d, e, f: Relevant geometry parameter distributions with the same color code as in Fig.2. **The solid black curves are obtained from the harmonic approximation of the wavefunction, while the colored ones are calculated from the full anharmonic wavefunction. Green curves present the maximum shifted towards longer bond lengths (Anharmonic increase), while the blue one stands for broadening of the anharmonic distribution with respect to the harmonic one and without maximum shift (Anharmonic broadening).** Specifically, panel b: Bond-length distributions along the radial O1-H6 stretch distance. The filled area is the histogram of the classical distribution obtained from the quasi-classical trajectory used to generate the semiclassical wavefunction. The harmonic normal mode distribution is small but not equal to zero at equilibrium distance because the variation of O-H stretch normal mode does involve, even if minimally, also other atom displacements. Panel c: Difference between anharmonic and harmonic curves of panel b. The statistical error bars for the distributions are smaller than the line width. Panel d: C2-C1-N angle distribution. Panels f, e: enlargement of the distribution peaks for the C2-H4/5 and C2-N bond distances. Panel g: Summary of the overall effect of anharmonicity and couplings on the quantum nuclear density which leads to specific nuclear density redistribution (indicated by the green arrows) giving effects which counterbalance each other in two separate regions of the molecule, the hydroxyl (highlighted in magenta) and the $CH_2NH_3^+$ (highlighted in green).”

II. REPLY TO REVIEWER'S #2 REMARKS

Reviewer #2: The manuscript by Aieta et al. discusses a computational investigation on geometry and vibrational modes of protonated Glycine obtained through a quantum description of the nuclei that include anharmonic effects. The average geometry of the molecule is found to be slightly elongated with respect to classical results, and the vibrational OH stretching acquires (due to anharmonicity) components on other normal mode as well.

The computational achievement (based on a clever scheme developed previously by M. Ceotto's group) is certainly remarkable. What concerns me most is the value of the new insights obtained from such challenging calculations. It is stated that the presented results contradict the "group frequency hypothesis". I do not think this is the case: here it is shown that if anharmonicity is taken into account, the OH stretching has some degree of mixing with other modes (mostly NH). But this is barely surprising, one of the expected effects of anharmonicity is to mix modes that are independent at the harmonic level (and mixing modes that are the closest in energy is the most natural occurrence). Already at textbook level the concept of group frequency is understood to be a useful but oversimplified picture of reality, the present work confirms this view.

Authors: We thank the Reviewer for their detailed analysis of our MS. We understand their concern about our claims on the "group frequency hypothesis". Their comments help us to improve our manuscript.

We agree with the Reviewer that it is obvious that beyond the harmonic approximation normal modes are mixing. However, as far as we know, no method at the present can calculate for moderately large molecules as GlyH⁺, and using ab initio potentials, the full-dimensional vibrational eigenfunctions for an arbitrarily vibrational level, proving not only which modes are mixing but also how much. This MS deals with these issues. Several other methods can go beyond the harmonic approximation, but most of them are focused on the calculation of the energy levels or related thermodynamics quantities. More specifically, apart from ground vibrational state distributions calculated with Monte Carlo methods, going beyond the harmonic approximation for excited vibrational eigenfunction calculations is still an open problem. Once one holds the eigenfunctions, one can calculate quantum mechanically all sorts of physical observables and has a deep physical picture into each mode coupling. We find in the literature that a frequent criterion is that normal modes involving nearest neighbor atoms should be mostly coupled (see for example X. Cheng and R. P. Steele, JCP 141, 104105 (2014); doi: 10.1063/1.4894507 and C. R. Jacob and M. Reiher, J. Chem. Phys. 130, 084106 (2009); <https://doi.org/10.1063/1.3077690>). The Reviewer is correctly suggesting another criterion according to which modes with similar energies are mostly coupled, including the overtones of lower frequency modes which are often coupled to higher frequency ones, as in the case of Fermi resonances. In a few words, the novelty of this MS is that it provides a practical solution for the calculation of the mode mixing from semiclassical dynamics on ab initio potential energy surfaces or on-the-fly, and for moderately large molecules, and the determination of how much intense the mixing is. And the value of the new insight of our results, that the Reviewer is wondering about in their question above, is that the eigenfunctions provided by our semiclassical method can tell which mode is coupled to which and how much, as the Reviewer is also pointing out below. In addition, semiclassical eigenfunctions can provide in principle any physical nuclear observable average that one may be looking for. When we put our findings into contrast with the traditional "group frequency hypothesis", for instance in this sentence in the abstract - "*More importantly, we introduce a new molecular interpretation of spectral peaks in vibrational spectroscopy showing that spectroscopic peaks are originated from the cooperation of several functional groups involving the whole molecular structure, contrary to the traditional "group frequency hypothesis"*" - we meant precisely the same concepts that the Reviewer wrote above. However, from the Reviewer's comments, we now realize that our phrasing may be read as exaggerated claims. What we meant in our claims is that this MS shows how to refine in a precise and quantitative way the textbook concept of group frequencies. In conclusion, we do agree with the Reviewer that it is well known already at the textbook level that the concept of group frequency is an oversimplified picture. Here we want to provide a possible computational tool to go a step forward the "group frequency hypothesis". The novelty of this MS is that our results represent a quantitative measure of the deviation from the corresponding harmonic estimates, not only for the vibrational energies but also for any other physical observable, because it can calculate the nuclear densities. In the light of these considerations, we follow the Reviewer's suggestion to soften some claims in the MS and better communicate the actual novelties of our work as described below or by looking at the markup version of the resubmitted MS.

In details, we modify the title from "*Anharmonic quantum nuclear densities from full dimensional vibrational eigenfunctions: an unexpected picture for protonated glycine vibrations*" to "*Anharmonic quantum nuclear densities from full dimensional vibrational eigenfunctions: an application to protonated glycine*".

Also, in the Abstract we change the sentences:

"More importantly, we introduce a new molecular interpretation of spectral peaks in vibrational spectroscopy showing that spectroscopic peaks are originated from the cooperation of several functional groups involving the whole molecular structure, contrary to the traditional "group frequency hypothesis". For example, several modes, in particular those

belonging to the amino group, contribute in a synergistic way to the intense O-H stretching signal, usually identified by the localized O-H stretch normal mode.”

as

“More importantly, our method is able to assign each spectral peak in vibrational spectroscopy by showing quantitatively how normal modes involving different functional groups cooperate to originate that spectroscopic signal. For example, several modes, in particular those belonging to the amino group, contribute in a synergistic way to the intense spectroscopic signal at 3,551 wavenumbers in the experimental IR spectrum. This is usually assigned almost exclusively to the O-H stretch normal mode.”

In the “Normal mode *vs* quantum anharmonic description of the O-H stretch excitation” Subsection we modify the sentence “More importantly, this picture highlights the relevant and unexpected coupling of the hydroxyl group with the far protonated amino group, which is significantly involved in the O-H stretch vibrational excitation.”

as:

“More importantly, the anharmonic nuclear density picture highlights the relevant coupling of the hydroxyl group with the far protonated amino group, which is significantly involved in the O-H stretch vibrational excitation”

In addition, in the “Discussion” section we rephrase the sentences

“This is quite counter-intuitive, given the position of the two functional groups in the molecule, and the normal mode picture that we are accustomed to. Instead, our semiclassical method can provide a quantum mechanical anharmonic picture of molecular vibrations where the traditional paradigm, according to which “one peak corresponds to one mode”, is replaced by a more comprehensive evaluation where more than one mode can be responsible for the same fundamental excitation and functional groups cooperate in vibrational excitations.”

as

“Even if the mixing of normal modes is the well known consequence of the introduction of couplings, the determination of the amount of mixing as resulting from our calculations is not trivial, also given the position of the two functional groups in the molecule, and the normal mode picture that we are accustomed to. Instead, our semiclassical method provides a quantum mechanical anharmonic picture of molecular vibrations which reveals quantitatively that more than one mode can be responsible for a given fundamental excitation and how much the modes are mixing. When we consider the complete potential, which includes all the anharmonicities without any other approximation apart from the level of electronic structure theory adopted, our calculations quantitatively characterize the involvement of other functional groups, also in the case of spectral features which are usually assigned to very localized normal modes, such as the O-H stretch one of the GlyH⁺.”

Also, the Reviewer is providing another important feedback when commenting about our “challenging” calculations. From their comment, we also understand that it is important to report the computational cost of our calculations. We think that our calculations are quite computationally cheap if compared with other quantum mechanical approaches. Our semiclassical calculations are based on on-the-fly evolution of just a single classical trajectory, one for each vibrational state. In this case one trajectory is for the ground state and another for the O-H stretch excited state. For each one, the Hessian matrix is calculated at each time-step. For our DFT simulations the timing is: 16h for the trajectory evolution and 552h for the Hessians employing the NWCHEM code on 20 CPUs with clock frequency of 2.6GHz. We also point out that the calculation of the Hessian along the trajectory is embarrassingly parallel, and the scaling with the number of cores employed is linear. The evaluation of wavefunction coefficients and the calculation of the densities takes few minutes on a personal computer, i.e. it does not necessarily require many cores or large memory. We added this information in the MS Section titled “Methods”.

Reviewer #2: In terms of quantify the extent of the effect, pictures such as fig. 3a may be misleading: they give the impression that OH stretching and other modes are mixed with comparable weights, but these are only the anharmonic nuclear densities on top of the harmonic ones. If the total (harmonic+anharmonic) density would be plotted, I expect a less impressive mixing, with the isosurfaces on NH rather smaller than those on OH.

Authors: In our terminology we define “harmonic” the calculations done under the normal mode approximation. Conversely, we use the term “anharmonic” to indicate the anharmonic calculations done with the actual potential at the DFT level of theory, i.e. without any additional approximation except from the semiclassical dynamics. Therefore, our anharmonic results are not provided as a correction term to be added to the harmonic ones, but they are already the final eigenfunctions, which includes both the harmonic and its anharmonic corrections. In the specific case of Fig.3a in the MS, we plot the difference between the anharmonic and harmonic nuclear densities. Thus, as the Referee says, that picture can be interpreted as the “correction” to the harmonic approximation to get the correct anharmonic molecular total density. As a further exemplification and to better reply to the Referee’s question, in the Fig.R3, we report some isodensities for the total vibrational density of the ground and excited vibrational states.

Figure R3. Plot of the nuclear density for different isosurface values. The first column identifies the vibrational eigenstate. Harm-ZPE stands for the harmonic approximated density for the ground eigenstate, while AnH-ZPE the corresponding anharmonic one. Harm-27 is the harmonic approximated density for the first excited vibrational eigenstate, and AnH-27 the corresponding anharmonic one.

The anharmonic densities are equivalent, using the Referee's terminology, to the total harmonic+anharmonic density. The total density has isosurfaces of comparable intensity and width on the NH_3^+ and on the OH functional groups. We would like to point out that the density does not give direct information about the anharmonic couplings. However, it does show the larger quantum delocalization of the lightest nuclei. We can easily spot any sort of mode coupling by looking at the density differences obtained by subtracting the harmonic density from the total anharmonic density, i.e that one obtained by accounting for the full potential, as reported in Figs. 2b, 3a, and 4a-d in the MS. Since we find this Reviewer's comment important and valuable, we decided to include these plots in our SI section as part of a more comprehensive picture. We suggest to look at the total anharmonic density plots when describing Fig.2 and Fig.3 in the revised MS on page 11 and page 13.

Reviewer #2: Concerning the ground state geometry modifications, again the results are quantitatively non trivial, but qualitatively expected: the asymmetry in bonding potentials, such as in Morse's, leads to change in the average bond length, even at the level of vibrational ground state.

Authors: We agree with the Reviewer that qualitatively a change was expected in the average bond lengths between harmonic approximation and anharmonic potential. We think the new insight provided by our method is precisely the quantitative assessment of these effects for a non-trivial system, as the Reviewer pointed out as well. It

is very interesting to note that the changes are smaller than one may have expected. More specifically, the harmonic geometry corresponds to the bottom of the potential well energy, i.e. the classical minimum. The quantum mechanical geometry provided by our eigenfunction corresponds to the zero point energy. In the case of protonated glycine this energy difference is more than 20,000 wavenumbers. In other words, the quantum mechanical ground state is 20,000 wavenumbers higher in energy with respect to the bottom of the well. In addition, with our calculations we are able to provide the nuclear densities of excited vibrational states, and thus also the average geometries for these states, a task that is not trivial for protonated glycine, and which is the main limitation of other existing techniques. Of course, in this case we observe geometry deviations which are more pronounced than the ground anharmonic ones from the bottom of the well estimates.

Reviewer #2: There are also some specific issues:

1. Computational procedures such as those in ref. 59 for glycine have been used to provide accurate molecular geometries. It is unclear to me how the anharmonic corrections found here should be integrated in such schemes: are they additional corrections to be accounted for in such protocols?

Authors: We thank the Reviewer for pointing out this issue. As anticipated above, we calculate the expectation geometry from the value of each operator averaged over the vibrational eigenfunction calculated from the anharmonic potential. More specifically, as explained in the SI of our MS, we calculate the full quantum distribution of the geometry parameters (bond length, bond angles and dihedrals), using the nuclear densities both in the harmonic and anharmonic case, and even for the excited states. The collection of this information is indeed the quantum expectation geometry of the molecule when the full potential (in the DFT approximation) is considered. Instead, in ref.59 another approach is taken, where the classical geometry, i.e. the bottom of the well, is corrected by appropriated perturbative terms. In our approach, there are no perturbative terms or corrections with respect to the harmonic ones, but the full anharmonic eigenfunction is calculated directly using the ab initio potential. Thus, we can not see at the moment how we can implement our anharmonic density into ref.59 perturbative scheme.

Reviewer #2: 2. From the supporting material, the procedure seems to provide also the coefficients of mixing of the various normal modes into the vibrational eigenfunction: could these be used to quantify the mixing of the OH stretching with the other modes?

Authors: This is a very good point and the Referee is correct. The starting point to calculate the nuclear densities are the vibrational eigenfunctions which we represent as an expansion on a harmonic basis set $|e_n\rangle = \sum_{\mathbf{K}} C_{n,\mathbf{K}} |\phi_{\mathbf{K}}\rangle$. The coefficients $C_{n,\mathbf{k}}$ quantify the weight of each K-esime harmonic state into the n-esime anharmonic vibrational state.

We report in Table RVII the coefficients in the case of the anharmonic excited O-H stretch state.

Table RVII. The first nine larger (in modulus) expansion coefficients for the anharmonic excited O-H stretch wavefunction. The most important contribution is given by the first row which corresponds to one quantum excitation of mode 27 (the O-H stretch one) and zero excitation for all the other modes.

K	k ₁	k ₂	k ₃	k ₄	k ₅	k ₆	k ₇	k ₈	k ₉	k ₁₀	k ₁₁	k ₁₂	k ₁₃	k ₁₄	k ₁₅	k ₁₆	k ₁₇	k ₁₈	k ₁₉	k ₂₀	k ₂₁	k ₂₂	k ₂₃	k ₂₄	k ₂₅	k ₂₆	k ₂₇	$C_{n=27,\mathbf{K}}$		
0	0	0	0	0	0	0	0	0	0	0	0	0	0	0	0	0	0	0	0	0	0	0	0	0	0	0	0	1	-6.09E-01	
1	0	0	0	0	0	0	0	0	0	0	0	0	0	0	0	0	0	0	0	0	0	0	0	0	0	0	0	1	0	3.53E-01
0	0	0	0	0	0	0	0	0	0	1	0	0	0	2	0	0	0	0	0	0	0	0	0	0	0	0	0	0	0	2.48E-01
0	0	0	0	0	0	0	0	2	0	0	0	0	0	0	0	0	0	0	0	0	1	0	0	0	0	0	0	0	0	-1.77E-01
0	0	0	1	0	0	0	0	0	0	0	0	0	0	0	0	0	0	0	0	0	0	0	1	0	0	0	0	0	0	-1.44E-01
0	0	0	0	0	0	0	0	0	0	0	0	0	0	0	0	0	0	0	0	0	0	0	0	0	0	0	0	2	1.29E-01	
0	0	0	0	0	0	0	0	1	0	0	0	0	0	2	0	0	0	0	0	0	0	0	0	0	0	0	0	0	0	-1.20E-01
0	0	0	1	0	0	0	0	0	0	0	0	0	0	0	0	0	0	0	0	0	0	1	0	0	0	0	0	0	0	-1.11E-01
0	0	1	0	0	0	0	0	0	0	0	0	0	0	0	0	0	0	0	0	0	0	0	0	0	0	1	0	0	0	1.02E-01

As one can see, the main contribution comes from the direct product of 27 harmonic eigenfunctions where only the harmonic O-H state is set at the first excited configuration and all the others at the ground one, as expected. However, in the latter case, there are other important contributions coming from other harmonic states that are mixing with

the fundamental O-H stretch one. For example, mode 26 which is the asymmetric N-H stretching at fundamental frequency equal to $3,504\text{ cm}^{-1}$, mode 25, which is the symmetric N-H stretching at frequency $3,445\text{ cm}^{-1}$, and modes 23 and 22 which are the C-H symmetric and asymmetric stretching with harmonic frequencies equal to $3,116\text{ cm}^{-1}$ and $3,105\text{ cm}^{-1}$ respectively form combination states with low frequency modes that have large coefficients in the anharmonic wavefunction expansion. Also, there is a contribution from the overtone of the O-H stretching with two quanta of excitation.

The coefficients for the anharmonic ground state are reported in Table RVIII. Here the contribution from the harmonic ground state is predominant.

Table RVIII. The first 9 largest (in modulus) expansion coefficients for the anharmonic ground state wavefunction. The most important contribution comes by far from the direct product of one-dimensional harmonic ground states reported in the first row, i.e. $k_i = 0$ for $i = 1, \dots, 27$.

k	k_1	k_2	k_3	k_4	k_5	k_6	k_7	k_8	k_9	k_{10}	k_{11}	k_{12}	k_{13}	k_{14}	k_{15}	k_{16}	k_{17}	k_{18}	k_{19}	k_{20}	k_{21}	k_{22}	k_{23}	k_{24}	k_{25}	k_{26}	k_{27}	$C_{n=1,k}$
0	0	0	0	0	0	0	0	0	0	0	0	0	0	0	0	0	0	0	0	0	0	0	0	0	0	0	0	9.28E-01
0	0	0	0	0	0	0	0	1	0	0	0	0	0	0	0	0	0	0	0	0	0	0	0	0	0	0	0	9.82E-02
0	0	0	0	0	0	0	0	0	0	0	0	0	0	0	0	0	0	0	0	0	0	1	0	0	0	0	0	-8.80E-02
2	0	0	0	0	0	0	0	0	0	0	0	0	0	0	0	0	0	0	0	0	0	0	0	0	0	0	0	8.09E-02
0	0	1	0	0	0	0	0	0	0	0	0	0	0	0	0	0	0	0	0	0	0	0	0	0	0	0	0	-7.68E-02
0	0	0	0	0	0	0	0	0	0	0	0	0	0	0	0	0	0	2	0	0	0	0	0	0	0	0	0	6.70E-02
0	0	0	1	0	0	0	0	0	0	0	0	0	0	0	0	0	0	0	0	0	0	0	0	0	0	0	0	-6.42E-02
0	0	0	0	0	0	0	0	0	0	0	0	0	0	0	0	0	0	0	0	0	0	0	1	0	0	0	0	5.31E-02
2	0	0	0	0	0	0	0	0	0	0	0	0	0	0	0	0	0	0	0	0	0	0	0	0	0	1	0	-4.78E-02

We add these tables with the eigenfunction coefficients into the SI and introduce these considerations about the quantification of the normal modes mixing into the “Results” section of the revised MS on pages 7 and 8.

Reviewer #2: 3. Fig. 3b: The harmonic density distribution of the first vibrational excited state of the OH stretching has no node. Why is that so?

Authors: Fig.3b in the MS shows the harmonic and anharmonic bond length density distributions. The Reviewer noted that the harmonic profile is not showing a node along the stretching direction but it presents a well-shape with its minimum just above zero. We start by observing that the harmonic O-H stretch normal mode displacement is quite localized along the O-H stretch direction, as one would expect. However, the mode is not completely local, as one can see from Table RIX, where the displacements of all atoms in Cartesian coordinates are reported at the turning point for a ZPE energy level.

Table RIX. Atomic displacements in Cartesian coordinates x , y , z for the maximum elongation (turning point) of the O-H stretch mode motion. The energy is set at the ZPE value of the mode. $|\mathbf{d}|$ is the displacement intensity, i.e. the squared root of the sum of the square of each component. Color code is red for the larger displacements, orange for the intermediate ones, and yellow for the small ones. The other displacements are negligible. The nuclei labels are the same as in Fig. 2a of the main text.

Harmonic Displacements (\AA)				
Nuclei	x	y	z	$ \mathbf{d} $
N	-0.00004	-0.00002	0.00000	0.00005
C2	-0.00004	-0.00005	0.00000	0.00006
H4	-0.00007	0.00017	-0.00015	0.00024
H5	-0.00007	0.00017	0.00015	0.00024
C1	0.00019	0.00005	0.00000	0.00019
O2	-0.00006	0.00009	0.00000	0.00011
O1	0.00521	0.00407	0.00000	0.00661
H3	0.00025	0.00028	0.00000	0.00037
H1	0.00013	0.00001	-0.00021	0.00024
H2	0.00013	0.00002	0.00021	0.00025
H6	-0.07225	-0.05759	-0.00005	0.09240

In particular, in the first column of this table, atoms O1 and H6 are those ones representing the O-H functional group into question and are highlighted in bold. Their displacements is by far larger than other atoms. However, also other atoms are involved, even if in a small amount, into this normal mode motion and that is the reason why we see that the density is not exactly zero, as one would expect. To recall a previous Reviewer’s comment, this is an example of how the “functional group hypothesis” is an approximation already at the harmonic level. In conclusions, the harmonic distribution for the O-H bond length reported in Fig.3b in the MS it would have shown a node only if the mode was completely localized along the O-H stretch. However this is not exactly the case because the O-H stretch normal mode involves also small atomic displacements of atoms either than H6 and O1. Therefore, we observe a density profile which is small in the middle of the distribution but not exactly zero. We added these considerations in the caption of Fig.3 in the MS.

Reviewer #2: 4. Fig. 2m: the pictorial representation using arrows of the anharmonic density plotted in fig. 2b is certainly useful. However I cannot understand why arrows on H4,H5 and H6 are missing, since there is anharmonic density also on these atoms.

Authors: We recall that the usage of the arrows in Fig.2m of the MS has the goal to express in a pictorial way the density shift due to the accumulation and depletion of the nuclear density when the complete full anharmonic potential, and the related density, are considered. As the Referee correctly writes, the arrows indicate the direction of the enrichment of the density for each nucleus separately. Instead, the colored glows indicate the overall main effects on the molecular geometry, i.e. the most marked molecular distortion with respect to the equilibrium one at the bottom of the potential which are deduced by the bond length, angles and dihedral distributions comparison. Therefore, as suggested by the Reviewer, we modify Fig.2m in the MS by adding the corresponding arrows at H4, H5 and H6 nuclei. We did not include them in the first place because these density shifts are less significant in the context of the overall molecular distortions with respect to the ones of other nuclei. In the case of the OH group, one can see from Fig.2b in the MS the depletion density red lobes on the H6 and O1 atoms on the same side. As a result, the O-H bond is not elongated when considering the anharmonic expectation geometry with respect to the harmonic one.

Figure R4. Enlargement of the maxima for the bond length and angle distributions involving the H4, H5, and H6. Continuous lines are the harmonic distributions, while dashed lines are the anharmonic ones.

One can also appreciate this from the bond length distribution reported in Fig.R4, where no significant shift can be observed with respect to the harmonic one (see Fig.R4a). Also the C1-O1-H4 angle is not affected by the inclusion of anharmonicity (see Fig.R4b). In the case of the C2-H4/5 bond, we observe a small shift of the maximum in the bond distance distribution (Panel c), of the same order of magnitude of the one in Fig.2c in the MS. Instead, we do not observe any significant change for the C1-C2-H5 bond angle, as represented in Fig.R4c. Therefore, we add also a green glow on the two CH bonds which result slightly elongated, just like the backbone of the molecule, and we also add the curve that we reported here in Fig.R4c as a new panel in Fig.2 of the MS, and change the caption accordingly. Instead, we choose, for the sake of brevity, to not show in the MS all the distributions that do not have significant shift with respect to the corresponding harmonic one.

Reviewer #2: In summary: I find the theoretical and computational achievement remarkable, the idea of representing the geometry and vibrational modes by real space nuclear densities effective and the new quantitative insights useful. Overall, all these points might warrant the broad interest expected for a Nature Communications paper. On the other hand, I think that the article in its present form is making too strong and unjustified claims, for the reasons I discussed above.

Authors: We thank the Reviewer for the supporting comments and we convey with them that the article in its past form presented too strong and unjustified claims. Following their suggestions, we modify the MS and rephrased our claims to rather convey the methodological power of our method, as already reported above and shown in the marked up version of the resubmitted MS file.

REVIEWERS' COMMENTS:

Reviewer #1 (Remarks to the Author):

The authors have addressed all comments by the reviewers and improved the manuscript. I believe that the manuscript can now be published in Nature Communications.

Reviewer #2 (Remarks to the Author):

I went through the authors' response and revised manuscript. All my concerns have been properly addressed, and I'm now in favour of publication in the present form. I've just spotted one typo (p.19 of the annotated version, in Method: "excitation state" → excited state), and perhaps in the abstract "signal at 3,551 wavenumbers" reads better as "signal at 3,551 cm^{-1} ".